# Extraordinarily long duration of Eocene geomagnetic polarity reversals
Yuhji Yamamoto [1] ✉, Slah Boulila[2,3], Futoshi Takahashi[4] & Peter C. Lippert [5]

One of the most intriguing attributes of Earth's magnetic field is that it reverses polarity. Previous palaeomagnetic records mainly from the last ~17 million years indicate that the reversal process typically occurs over ten thousand or fewer years. Here we present an exceptionally resolved deep sea sedimentary palaeomagnetic record that captures two polarity transitions that occurred ~40 million years ago. These transitions occur over 18 and 70 thousand years and are much longer than the typical 10 kyr duration for younger reversals. Longer-duration transitions like these are calculated in numerical geodynamo models, suggesting that variable reversal durations are an intrinsic property of the geodynamo. This variability predicts that polarity transition durations can be much longer than 10 kyr. Occasional prolonged intervals of transitional, and weaker, geomagnetic fields would have exposed Earth's Eocene environments to greater high-energy radiation from the sun for longer intervals of time, with potential impacts on biota.

Earth's magnetic field is one of the most intrinsic properties of our planet. The field is generated by fluid motion of an iron-nickel outer core and may have started ~4.2 billion years ago[1]. The geomagnetic field changes its polarity stochastically and may have begun reversing ~3.5 billion years ago[2]. Dipole energy is drastically reduced during geomagnetic reversals[3,4], resulting in dynamic variations in both the field intensity and direction. Whereas volcanic records can provide precise, but temporally incomplete records of direction and intensity [5], marine sediments can capture continuous geomagnetic field variations[6] and provide complementary and fundamental information of the reversal process[7]. Sediment records for the last four reversals (back to 1.78 million years ago (Ma)) suggest that the entire reversal process in direction (stable polarities of opposite directions) takes approximately 1-12 thousand years (kyr) depending on the latitude at which the reversal is observed[8]. Semi-continuous palaeomagnetic records of seven field reversals obtained from sequential lava flows on land for two intervals from 0.78 to 16.7 Ma and at 180 Ma suggest that the reversal process comprises three successive and distinct phases of magnetic direction and dipole intensity: a precursor phase lasting ~2.5 kyr; a main transition phase lasting ~1 kyr; and a rebound phase lasting ~2.5 kyr[9]. These records also indicate that the total reversal duration in direction is ~9 kyr[9]. The most detailed sedimentary record of the last geomagnetic reversal (Brunhes-Matuyama Transition, ~0.77 Ma) dates the reversal duration in direction to ~11 kyr[10]. Thus, it is commonly assumed that geomagnetic reversals in

direction last ~10 kyr from start to finish based on what we observe and measure from reported palaeomagnetic records.

Approximately 540 geomagnetic reversals have occurred over the last ~170 million years[11]. The estimated ~10 kyr duration of the entire reversal process in direction deduced from just seven reversal records[8,9], which only represent <1.3% of reversals over the last ~170 million years. Thus, this duration estimation may not represent intrinsic geomagnetic behaviour. For example, some numerical geodynamo simulations (without data assimilation) suggest that reversal durations follow a log-normal law, thus permitting more temporal variability and longer reversal durations than commonly assumed[12]. Although geodynamo simulations do not perfectly mimic Earth-like dynamos, they nonetheless provide time-series data that are remarkably similar in behaviour to the Brunhes-Matuyama Transition[13].

Here we present a highly resolved sedimentary record of two Eocene reversals that occurred ~40 million years ago, which have extraordinarily long durations. Although numerous sedimentary palaeomagnetic records have been reported, high-resolution sedimentary records like those reported here are rare. Previously, the oldest similarly high-resolution sedimentary records of geomagnetic reversals were dated to ~2.58 Ma[14] and ~3.33 Ma[15,16], giving directional reversal durations of ~10 kyr. We compare our observations of Eocene data to models from numerical geodynamo simulations to further explore directional reversal duration variability.

[1]Marine Core Research Institute, Kochi University, Kochi, Japan. [2]Sorbonne Université, CNRS, Institut des Sciences de la Terre Paris, ISTeP, Paris, France. [3]LTE, CNRS, Observatoire de Paris, PSL University, Sorbonne Université, 77 avenue Denfert-Rochereau, Paris, France. [4]Department of Earth and Planetary Sciences, Faculty of Science, Kyushu University, Fukuoka, Japan. [5]Department of Geology & Geophysics, University of Utah, Utah, USA. ✉ e-mail: y.yamamoto@kochi-u.ac.jp

## Results and Discussion

### Middle Eocene palaeomagnetic polarity transition record

Integrated Ocean Drilling Program (IODP) Expedition 342 recovered Palaeogene sedimentary drift sequences from the Newfoundland ridges in the northwest Atlantic Ocean[17]. The studied Site U1408 sediments were deposited at average rates of ~2.4 cm/kyr and have prominent lithological alternations that can be detected robustly from multiple chemical and physical proxy data (Fig. 1)[18]. We used scanning X-ray-fluorescence-measured Ca/Fe ratios to calibrate the geochronology of most of the middle Eocene sediments, based on the stable 173-kyr-obliquity cycle of Earth's axial tilt amplitude modulation. The robust geochronology helps to elucidate relative palaeointensity (RPI) variations of the geomagnetic field during the middle Eocene (between ~38.4 and ~43.2 Ma), although the signal is smoothed because measurements were made on continuous sample sections[19]. We reanalyzed the polarity transition intervals at the beginning and end of Chron C18n.1r using discrete samples collected approximately each 2 cm over 8 m of section from Holes A and C at Site U1408. Results from discrete samples avoid smoothing artifacts in the time series of direction and intensity change and allow more precise palaeo- and rock magnetic measurements.

Measurements of the natural remanent magnetization (NRM) reveal a characteristic remanent magnetization (ChRM) component after removal of a secondary remanent magnetization typically by 20 mT (Extended Data Fig. 1). Our rock magnetic measurements indicate that the remanent magnetizations are carried mainly by primary biogenic magnetite, as evidenced by Verwey transitions at ~100 K[20,21] and distinctive central-ridges in first-order reversal curve (FORC) diagrams[22–24] (Extended Data Fig. 2). Thus, the ChRM components represent primary palaeomagnetic directions from which we calculated the latitudes of the positions of the north virtual geomagnetic pole (VGP). We used measurements of the anhysteretic remanent magnetization (ARM) and isothermal remanent magnetization (IRM) to calculate ratios of NRM to

ARM (Extended Data Fig. 3) and NRM to IRM to evaluate RPI (Extended Data Fig. 4).

Downcore VGP and RPI variations from Holes A and C were calculated and plotted on the common depth scale for Site U1408 (Extended Data Fig. 5). Intervals with VGP latitude higher (lower) than +45° (−45°) with continuity was then assigned as a normal (reversed) polarity interval; other intervals are interpreted to record polarity transitions. Our cut-off angle of ±45° is a conservative criterion for defining magnetic polarity. A composite VGP and RPI record is shown in Fig. 2 with interpreted geomagnetic polarity chrons assigned based on the biostratigaphy and magnetostratigaphy of Site U1408[25,26]. Two polarity transition intervals are identified: the end of normal-polarity subchron C18n.2n to the onset of reversed-polarity subchron C18n.1r occurs over approximately a 50 cm interval (transition 1; Fig. 2b), and the end of C18n.1r to the onset of normal-polarity subchron C18n.1n occurs over an interval of 170 cm (transition 2; Fig. 2b). These transitions are associated with low RPI (Fig. 2c), which is typical of polarity transitions observed over the last ~50 million years[19,27–29].

The geochronology based on the 173-kyr obliquity amplitude modulation cycle[18] provided calibrated ages of 39.59-39.60 Ma for transition 1 and 39.43-39.50 Ma for transition 2 (Fig. 3a). Thus, deposition rates are estimated to be 2.4-2.8 cm/kyr for these intervals. If we adhere as closely as possible to the simplified model that the reversal process comprises three successive and distinct phases[9], then we recognize only a main transition that lasted for ~2 kyr and two rebound phases that lasted for ~11 and ~5 kyr for transition 1 (Fig. 3b). RPI remains low for ~38 kyr (Fig. 3c). All three reversal phases[9] are recognized in transition 2 (Fig. 3b): the precursor phase lasted ~22 kyr, the main transition occurred over ~9 kyr, and the rebound phase comprises three intervals with ~14 kyr, ~13 kyr, and ~16 kyr durations. RPI remains low for ~70 kyr (Fig. 3c). This simplified model of reversal behaviour[9] is based on two assumptions: first, that each polarity interval has the same duration, and second, that each polarity reversal has the same morphology, with the latter assumption relying on the former[30].

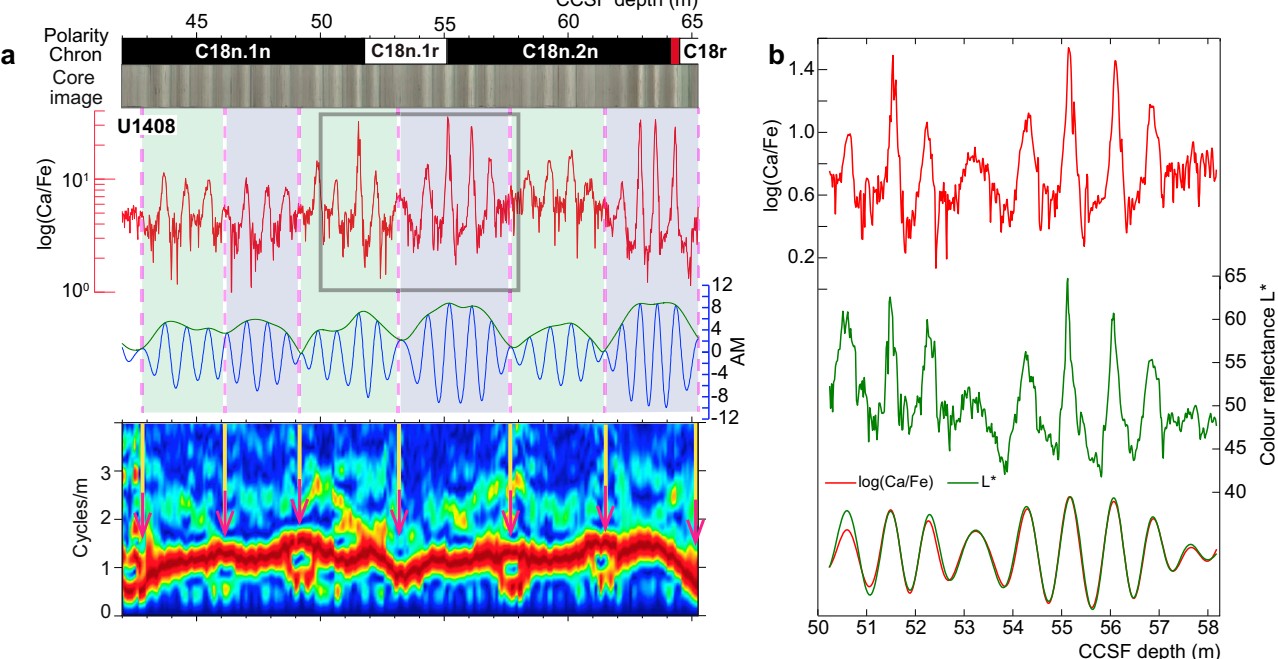

**Fig. 1 | Integrated stratigraphic data for IODP Site U1408[18]. a** Litho-magnetostratigraphic and scanning X-ray-fluorescence-measured Ca/Fe ratio data. Rhythmic lithological alternations expressed in sediment colour, where dark colours correspond to clay-rich sediments, and light colours to carbonate-rich sediments (see also '**b**'). The elemental lithological cyclicity (obliquity, ~40 kyr) is bundled by a longer cyclicity (~173 kyr) shown by vertical, dashed lines and detected by the amplitude modulation (AM) method and the amplitude spectrogram, both of which

were applied to highly resolved (2 cm intervals) XRF Ca/Fe ratio data. The spectral line in the amplitude spectrogram represents the elemental lithological cyclicity, and bifurcations within the spectral line (indicated by vertical arrows) track the longer ~173 kyr cyclicity. **b** Expanded view of the studied interval with XRF Ca/Fe and colour reflectance L* data, along with Gaussian bandpass (1.1 ± 0.5 cycles/m) filter outputs to extract the elemental cyclicity.

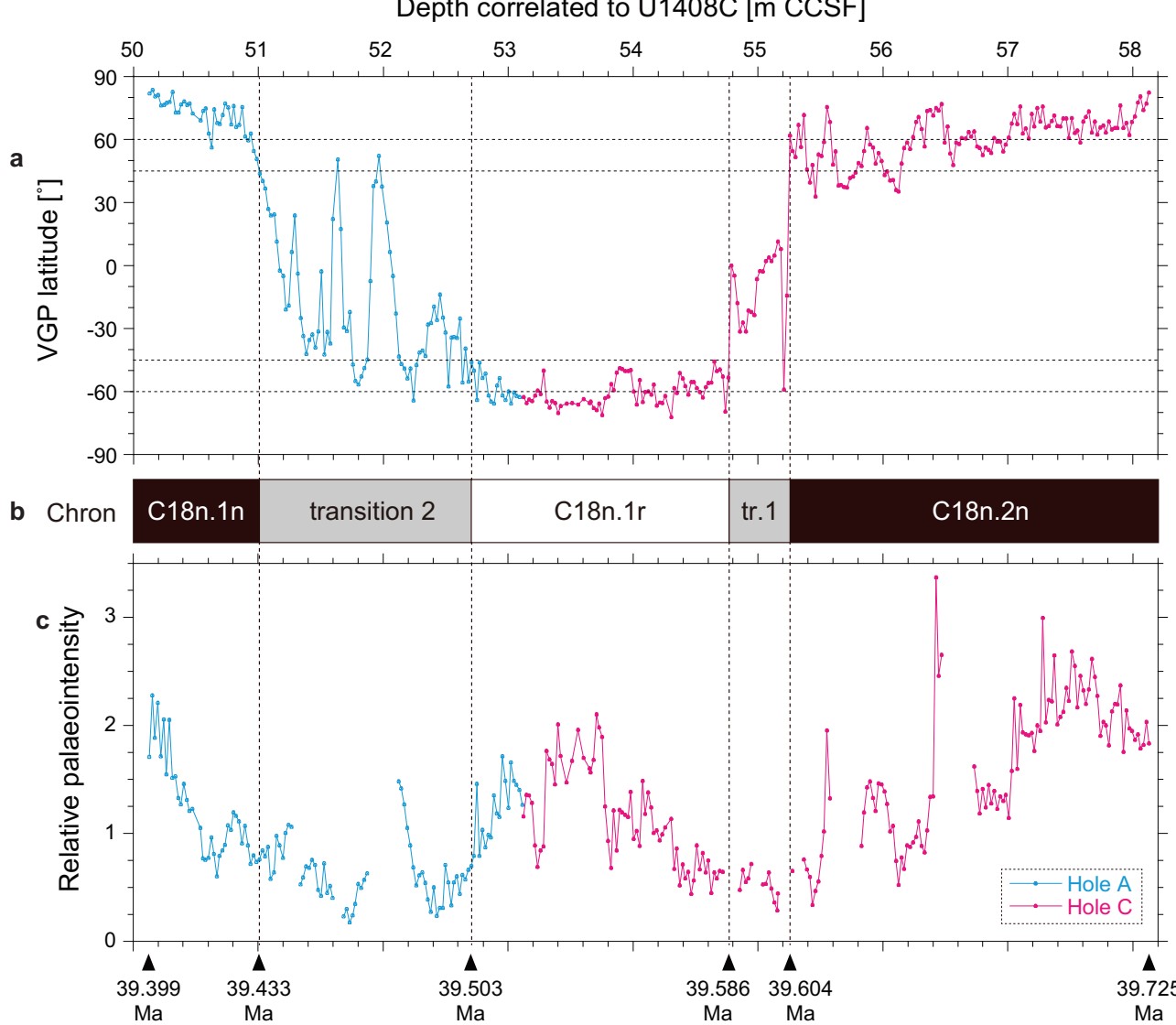

**Fig. 2 | Composite VGP and RPI record with interpreted geomagnetic polarity chrons.** Downcore variations of the virtual geomagnetic pole (VGP) latitude **a**, geomagnetic polarity chrons **b**, and the relative palaeointensity (RPI) of the geomagnetic field **c** from an approximately 8 m composite interval of sediment at IODP Site U1408.

The occurrence of multiple rebounds is not unprecedented: this behaviour is also reported for the Bruhnes-Matayama reversal[31]. We suggest that it may be more common and that polarity reversals are inherently complex, if not somewhat chaotic, events.

To assess uncertainties on durations of polarity chrons and their transitions, we used the *"Undatable"* age-depth modeling routine, which generates age-depth models including uncertainties in both age and depth for [14]C and calendar ages[32]. Based on the astronomical timescale[18], we used critical age-depth points for uncertainty estimates (Extended Data Fig. 6). We included errors on the depth scale resulting from our discrete sampling for palaeomagnetic data, along with age errors from variable sedimentation rates. The resulting duration of the entire reversal process in direction is estimated to be 18 ± 3 kyr for transition 1 and 70 ± 6 kyr for transition 2. These are much longer than the typical ~10 kyr duration based on published palaeomagnetic records.

### Variability of reversal durations from numerical geodynamo simulations

The typical, but actually assumed duration of ~10 kyr for geomagnetic directional reversals is derived from a composite record of 7 geomagnetic reversals[8,9], or only ~1% of the reversals that have occured over the last ~180 million years. Numerical geodynamo simulations suggest that reversal durations are distributed log-normally[12], such that longer transition durations are possible and expected even if they are not yet observed widely. We evaluate this with our own numerical geodynamo model, which does not use data assimilation. Our model produced 160 reversal events. We define directional reversal durations by the interval during which the dipole axis stays within the ±45° latitude bounds[33,34]. This is a conservative definition that yields shorter reversal durations. Directional reversal durations vary among runs (Extended Data Table 1) with an average from 0.18 to 0.64 in the non-dimensional unit (Fig. 4). The upper bounds of these durations are between 0.82 and 3.30, which suggests that the longest reversal duration ranges from ~33 to 130 kyr when we use a dipole free decay time of ~40 kyr to rescale the non-dimensional time unit. We note that transition durations would even be longer if we used a higher electrical conductivity value for Earth's core[35], or a less conservative reversal definition. Time rescaling can be challenging, however, because of high viscosity in the present numerical models as well as in models in a previous study[12]. Time rescaling using the dipole decay time is done based on an assumption that the reversal duration is controlled by diffusive processes[12]. On the other hand, the secular-

**Fig. 3 | Reversal path phases for Chron 18n.1r and age variations of reversals.** Classifications of precursor (blue), transition (red), and rebound (green) are based on the literature[9] and age variations of reversals are based on the astronomically-tuned age model for Site U1408[18]. Geomagnetic polarity chrons **a**, VGP latitude **b**, and RPI **c**. Transitions are characterized by low RPI intervals (red horizontal lines in the RPI record).

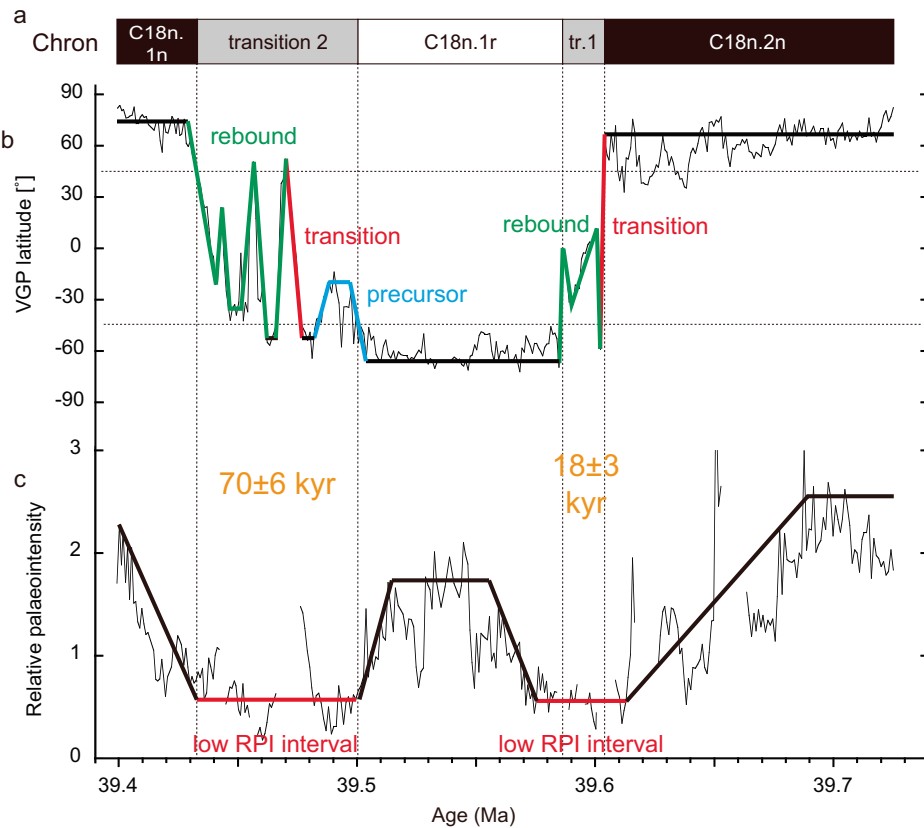

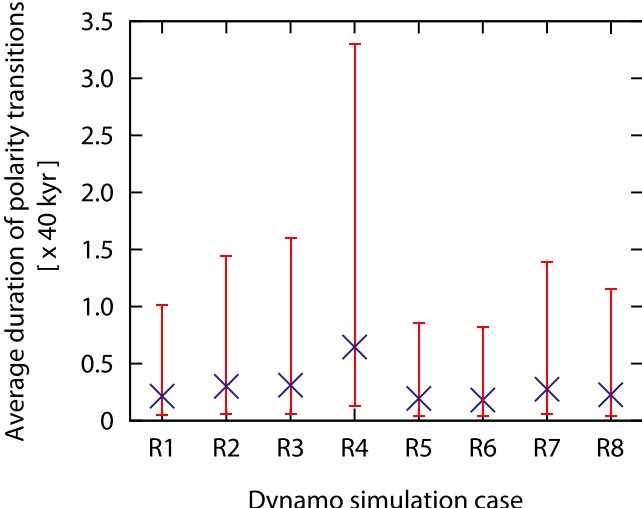

**Fig. 4 | Average durations of polarity reversal events from our numerical dynamo models.** Dynamo simulation cases R1 to R8 are indicated on the horizontal axis, which vary in their assigned parameter values of different Ekman and Rayleigh numbers, while the vertical axis represents the duration for each case; this value should be multiplied by 40 kyr to convert to approximate Earth years. The range of duration is represented by the 2-sigma-level intervals.

variation timescale, which is inversely proportional to the magnetic Reynolds number, could be used for rescaling time focusing on secular-variation processes[36]. In this case, a correction factor of 0.17–0.37 given by the ratio of the magnetic Reynolds number in the numerical models (167–372, Extended Data Table 1) to that in the Earth's core (~1000)[37] should be multiplied by the dipole decay time for time rescaling. This alternative rescaling approach yields shorter reversal durations.

The estimated 18 ± 3 and 70 ± 6 kyr durations for transition 1 and 2, respectively, are both admissible in view of these numerical dynamo model results. We cannot exclude uncertainty, bias, or both in calibrating time from dimensional units due to the vast differences in parameters between Earth's core and numerical dynamo models[37]. Results from numerical dynamos suggest that reversals with long durations are characterized by complex behaviours of the dipole axis, similar to our observations of field behaviour during transition 2, in which repeated multiple precursor, transitional, and rebound phases occur during a single field reversal (Extended Data Fig. 7). Although such behaviours are qualitatively similar among the reversals[13], mechanisms for such complex dipole variations remain unclear and we cannot yet deduce any physics-based constraints on typical timescales for directional reversal durations. More realistic and advanced dynamo models with relevant force balance[38] are necessary to explore what controls the reversal duration. However, it is still computationally impractical to use state-of-the-art dynamo models to obtain a sufficient number of reversals to assess their statistical properties, including reversal durations[39]. Thus, our approach of using modest viscosity parameters is one reasonable way to explore reversal duration. Effects of core-mantle boundary heat flux variation and/or a stable stratified layer at the top of the core, which are not considered in the present models, may play a role in controlling reversal durations, reversal frequency, and geomagnetic secular variation[33,40].

## Implications

Our findings suggest that variable reversal durations are an intrinsic geodynamo property. Moreover, this variability predicts polarity transitions that can be much longer than 10 kyr. Our Site U1408 is presently at ~41 °N; its palaeolatitute at the time of sediment deposition (assuming a time-averaged geocentric axial dipole (GAD) field structure) is estimated to be ~22 °N (measured inclination of ~40°)[26]. If the latitude dependence of reversal durations inferred from the last four reversals, namely, greater durations observed at higher latitudes[8], is also an intrinsic long-term geodynamo property, then reversal durations are expected to be even longer

than $70 \pm 6$ kyr at mid to high latitude regions for transition 2. Some research suggests that the Eocene geomagnetic field was characterized by non-GAD field components larger than the recent field, particularly larger octupole contributions[41] or by a more non-axial dipole[42]. Deviations from a GAD field are expected to underestimate the true latitude of reversal observation (the sampling site) and produce more non-typical reversal behaviour. Therefore, our results may be explained in part by non-GAD field contributions. Regardless, we observe two reversals that have the same directional structure as recent reversals and with durations that far exceed previous chronologies. Occasional prolonged intervals of transitional, and weaker, geomagnetic fields, with or without large non-GAD contributions, would have exposed Eocene environments to greater high-energy particle radiation from the sun for longer intervals of time such as in the Ediacaran[43,44]. This prolonged exposure may have influenced geochemical cycles, surface procceses, and biota of the Eocene. Testing such teleconnections merits further attention.

## Methods

### Samples

Discrete samples were collected onboard the R/V *JOIDES Resolution* during IODP Expedition 342 in 2012. Non-magnetic plastic cubes with a volume of 7 cm³ were inserted continuously into the central part of the working half cores with ~2.2 cm spacing. In total, 191 samples were collected from Hole A between 46.81 and 51.45 m CFS (core depth below sea floor) and 262 samples were collected from Hole C between 49.82 and 55.78 m CFS. A between-hole composite depth scale in metres CCSF (core composite depth below sea floor) was constructed based on between-hole correlation of physical, chemical, and magnetic properties[18,25]. Mid-point depths of the samples correspond to 50.17-54.80 m CCFS (Hole A) and 52.18-58.13 m CCFS (Hole C).

### Palaeo- and rock magnetic measurements

Each sample was first analyzed by measuring its natural remanent magnetization (NRM) and then was stepwise demagnetized using alternating field (AF) techniques to a peak field of 80 mT, typically involving 21 demagnetization steps. An anhysteretic remanent magnetization (ARM) was then imparted to each sample with a DC bias field of 100 μT and a peak AF of 80 mT and was then measured, followed by stepwise AF demagnetization of the ARM by a peak field of 80 mT. An isothermal remanent magnetization (IRM) was then imparted to each sample using a pulsed field of 2.5 T; we regard this value as the saturation IRM (SIRM) for each sample. This SIRM was then subjected to stepwise AF demagnetization to a peak field of 80 mT. Measurements of the remanences and stepwise AF demagnetizations were made using a 2 G Enterprises model 760 R cryogenic magnetometer system with an inline static AF demagnetizer and a Natsuhara-Giken model ASPIN-A spinner magnetometer. IRMs were imparted using a Magnetic Measurements model MMPM-10 pulse magnetizer.

After the remanence measurements, some samples were freeze-dried to conduct low-temperature magnetometry using a magnetic property measurement system (Quantum Design model MPMS-XL5) and first-order reversal curve (FORC) measurements using an alternating gradient magnetometer (Lake Shore model MicroMag 2900 AGM). Small fractions of the samples (few tens of mg) were used for the low-temperature magnetometry: the fraction was first cooled from 300 to 5 K in a 3 T field (3T-FC-remanence) and temperature variation of the 3T-FC-remanence was monitored from 5 to 300 K in zero field in 1-1.5 K steps. The fraction was again cooled from 300 to 5 K in zero field and then an IRM was imparted in a 3 T field (3T-ZFC-remanence), followed by measurement of the temperature variation of the 3T-ZFC-remanence from 5 to 300 K in zero field. An IRM was further imparted for the fraction in a 3 T field at 300 K (3T-SIRM) and variation of the 3T-SIRM was measured between 300 and 5 K. FORC measurements were performed on another small sample fraction using the following measurement parameters: 263 FORCs; field increment of 1 mT; local interaction field ($H_u$) between −50 and 50 mT; coercivity ($H_c$) from 0

to 150 mT; maximum applied field of 250 mT; averaging time of 100-150 ms for each data point. FORC diagrams[45] were produced using the FORCinel software with the VARIFORC algorithm to improve smoothing[46,47].

### Determination of virtual geomagnetic pole (VGP) latitude

Stepwise AF demagnetization results of NRM were used to calculate characteristic remanent magnetization (ChRM) directions of the samples using principal component analysis (PCA) with a fit anchored to the origin[48]. Except for a few samples, ChRM fits have maximum angular deviation (MAD) values less than 10° (Extended Data Fig. 1). The cores were recovered from the seafloor using the FlexIt core orientation tool, which provides the magnetic tool face (MTF) orientation value giving the angle between geomagnetic north and the double line on the core liner for each core[49]. Thus, the relative declination of the ChRM direction for each sample is transferred into a geographic coordinate to calculate a virtual geomagnetic pole (VGP) latitude.

### Determination of relative palaeointensity (RPI) of the geomagnetic field

Stepwise AF demagnetization results of NRM, ARM, and IRM were used to construct diagrams of NRM-ARM and NRM-IRM for each sample. Best-fit slopes were individually determined, giving values of the slopes as $slope_{NRM-ARM}$ and $slope_{NRM-IRM}$ (Extended Data Figs. 3 and 4). Linearity is assessed using the correlation coefficient (r) of the linear regression, which in most cases exceeds 0.95. To define the linear segment, we typically use a coercivity range from 15–25 mT (lower bound) to 70–80 mT (upper bound). This selection is made to avoid the influence of secondary magnetization, which is commonly carried by grains with coercivities below 15–25 mT, most probably detrital magnetic grains. As a result, the slopes are primarily derived from a coercivity interval of 20–80 mT, ensuring sufficient linearity and minimizing contamination from secondary components. Given that the higher coercivity portion is primarily controlled by the biogenic component[50,51], we infer that the slopes predominantly reflect palaeomagnetic records carried by the biogenic component.

Overall averages of $slope_{NRM-ARM}$ were calculated separately for samples from Hole A ($average_{NRM-ARM, Hole A}$) and Hole C ($average_{NRM-ARM, Hole A}$). They were used to normalize $slope_{NRM-ARM}$ for each sample from Hole A ($slope_{NRM-ARM}/average_{NRM-ARM, Hole A}$) and Hole C ($slope_{NRM-ARM}/average_{NRM-ARM, Hole C}$). The same calculations were made on $slope_{NRM-IRM}$ ($slope_{NRM-IRM}/average_{NRM-IRM, Hole A}$; $slope_{NRM-IRM}/average_{NRM-IRM, Hole C}$). Relative palaeointensity (RPI) of the geomagnetic field was estimated as an average of $slope_{NRM-ARM}/average_{NRM-ARM, Hole A}$ and $slope_{NRM-IRM}/average_{NRM-IRM, Hole A}$ for individual samples from Hole A, and by an average of $slope_{NRM-ARM}/average_{NRM-ARM, Hole C}$ and $slope_{NRM-IRM}/average_{NRM-IRM, Hole C}$ for individual samples from Hole C. Some samples have inhomogeneous magnetic properties represented by the ratio of ARM to SIRM (ARM/SIRM): RPI for samples with ARM/SIRM deviating from the average (0.184 for Hole A; 0.180 for Hole C) by a one standard deviation or more were discarded.

### Downcore VGP and RPI variations in the common depth scale

Palaeomagnetic measurements were taken from the lower portion of the core from Hole A and the upper portion from Hole C. Because core disturbances are typically found at both ends, we initially excluded data from the bottom ~0.3 m of Hole A and the top ~0.4 m of Hole C to avoid potential data contamination. When downcore VGP and RPI variations are plotted on the between-hole composite depth scale in metres CCSF, cm-scale offsets are evident between Hole A and Hole C for the overlapping interval (~52.2-54.8 m CCSF). Calibration of the geochronology was mainly made using the X-Ray fluorescence-derived Ca/Fe record from Hole C[18]. We held the Hole C depths fixed (CCFS$_{Hole C}$) and shifted the Hole A depth (CCFS$_{Hole A}$) with respect to the Hole C depth to match the variation between Hole A and Hole C. We found that a 4-cm upward-shift (CCFS$_{Hole A}$ − 0.04 m) produces the best match for variations between the two holes, so the Hole A depth was correlated to the Hole C depth (CCFS$_{Hole C}$) in this way (Extended Data

Fig. 5). RPI values from Hole C were rescaled to have the same average as those of Hole A for the overlapping interval. The composite VGP and RPI variations were adopted from the 50.13-53.12 m CCFS$_{Hole C}$ interval in Hole A and that of 53.12-58.13 m CCFS$_{Hole C}$ in Hole C.

The VGP variation was interpreted based on palaeomagnetic directions determined using PCA with a fit anchored to the origin. This approach generally yields lower uncertainty compared to an unanchored fit[52]. In our case, the anchored fit indeed resulted in lower uncertainties than the unanchored fit; however, the overall characteristics of the VGP variation remain consistent between the two methods (Extended Data Fig. 5a and 5b).

### Numerical geodynamo simulations

We solve the equations for chemically-driven convection of the Boussinesq fluid, transport of the light element concentration, and magnetic induction in a spherical shell rotating with an angular rotation rate, $\Omega$[53,54]. The ratio of the inner core radius ($r_i$) to that of the outer core ($r_o$) is set to $r_i/r_o = 0.35$. The inner core and the mantle are assumed to be insulating. Boundary conditions are no-slip for the velocity field, and fixed flux for the light element concentration. We set the ratios between the fluid viscosity $v$, the compositional diffusivity $\kappa^C$ and the magnetic diffusivity $\eta$ to give the compositional Prandtl number $Pr^C = v/\kappa^C = 1$ and the magnetic Prandtl number $Pm = v/\eta = 20$. The Ekman number is set to $Ek = v/2\Omega D^2 = 3.25 \times 10^{-3}$ and $2 \times 10^{-3}$ (here $D = r_o - r_i$ is shell thickness). The Rayleigh number $Ra = \alpha g \varepsilon D^3 / 2\Omega v \kappa^C$ (here $\alpha$ is the rate of compositional expansion, $g$ is the gravitational acceleration at the core-mantle boundary, and $\varepsilon$ is the uniformly distributed volumetric sink of light elements, respectively) is varied to obtain dynamo solutions.

We simulated eight dynamos with different $Ek$ and $Ra$ values; these are named R1 – R8, respectively. The start of a directional reversal is defined as the instance when the dipole axis crosses the latitude of ±45˚ toward the equator before polarity change. Similarly, the end of a reversal in direction is defined as the instance when the dipole axis finally crosses the latitude of ±45˚ toward the geographic pole after crossing the equator. Stability of the dipole axis after crossing the equator for at least one dipole free decay time is required to distinguish reversals from short events such as excursions[34,40]. The mean and standard deviation of the sampled reversal durations in cases R1 – R8 are calculated according to a lognormal distribution[13]. Our definition of a directional reversal is conservative, which results in shorter estimated durations than other definitions. Time is scaled using the dipole free decay time $\tau_{dip} = r_o^2/(\pi^2 \eta)$, or ~ 40 kyr using a conservative estimate of $\eta = 1$ m$^2$s$^{-1}$[55]. The latest estimate of higher electrical conductivity yields a longer dipole free decay time[35]. Properties of the dynamo solutions are summarized in Extended Data Table 1.

### Data availability

The data that support the findings of this study are available in the extended data table (key result values of numerical geodynamo simulations) and via Zenodo (https://doi.org/10.5281/zenodo.13917223).

### Code availability

The code used for the numerical geodynamo simulation is available via Zenodo (https://doi.org/10.5281/zenodo.13917223).

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

## Acknowledgements

This research used samples and data provided by the Integrated Ocean Drilling Program (IODP). We thank the staff of the R/V *JOIDES Resolution* and the Bremen Core Repository, and the Expedition 342 science party led by R.D. Norris and P.A. Wilson. This research was supported through the Japan Agency for Marine-Earth Science and Technology (JAMSTEC) IODP Expedition 342 After Cruise Research Program, the Japan Society for the Promotion of Science (JSPS) KAKENHI (JP15H05832, JP16H04043, JP18K03808, JP21K03725 and JP24K07119), and the Kochi University Research Project (Earth Investigation Project: Past, present, and future of environment, earthquake, and resources recorded in the ocean and land). S.B. was supported by the French Agence Nationale de la Recherche (19-CE31-0002 AstroMeso) and the European Research Council under the European Union's Horizon 2020 Research and Innovation Program (Advanced Grant AstroGeo-885250). Computation was performed using computer facilities at the Research Institute for Information Technology, Kyushu University. Constructive comments made by an anonymous reviewer and Andrew P. Roberts improved the manuscript.

## Author contributions

Y.Y., S.B. and P.C.L. designed the project. Y.Y. conducted palaeo- and rock magnetic measurements and analyzed the results. S.B. developed the geochronology model. F.T. conducted numerical geodynamo simulations and analyzed the results. All authors contributed to discussion of results and editing of the manuscript.

## Competing interests

The authors declare no competing interests.
