## [Transparent Peer Review file · Communications Earth & Environment]

Extraordinarily long duration of Eocene geomagnetic polarity reversals

Corresponding Author: Dr Yuhji Yamamoto

Version 0:

Decision Letter:

Dear Dr Yamamoto,

Your manuscript titled "Extraordinary long duration of Eocene geomagnetic polarity reversals" has now been seen by 2 reviewers, and we include their comments at the end of this message. They find your work of interest, but some important points are raised. We are interested in the possibility of publishing your study in Communications Earth & Environment, but would like to consider your responses to these concerns and assess a revised manuscript before we make a final decision on publication.

We therefore invite you to revise and resubmit your manuscript, along with a point-by-point response that takes into account the points raised. Please highlight all changes in the manuscript text file.

In the revised version, please make sure to:

- Fully clarify the assumptions on which the numerical simulations are based, giving particular attention to the estimation of the viscosity forces.
- Clarify and discuss the classification of the stages of polarity reversals based on the available literature.
- Provide estimation of uncertainties whenever possible and improve the presentation of the results, as suggested by Reviewer #2.

Please submit your point-by-point responses as a separate file, distinct from your cover letter where you can add responses to the Editors' comments that you do not want to be made available to the reviewers. Word files are preferred. We recommend that any figures, tables or graphs that are included in the response to reviewers are also included in the main article or Supplementary Information.

Please use the following link to submit your revised manuscript, point-by-point response to the referees' comments (which should be in a separate document to any cover letter), a tracked-changes version of the manuscript (as a PDF file) and the completed checklist:

Link Redacted

We hope to receive your revised paper within six weeks; please let us know if you aren't able to submit it within this time so that we can discuss how best to proceed. If we don't hear from you, and the revision process takes significantly longer, we

may close your file. In this event, we will still be happy to reconsider your paper at a later date, as long as nothing similar has been accepted for publication at Communications Earth & Environment or published elsewhere in the meantime.

Please do not hesitate to contact us if you have any questions or would like to discuss these revisions further. We look forward to seeing the revised manuscript and thank you for the opportunity to review your work.

Best regards,

Joao Duarte, PhD
Editorial Board Member
Communications Earth & Environment
orcid.org/0000-0001-7505-3690

Alireza Bahadori, PhD
Senior Editor
Communications Earth & Environment
Consulting Editor
Communications Sustainability

EDITORIAL POLICIES AND FORMATTING

- Behavioural and social science
- Ecological, evolutionary & environmental sciences
- Life sciences

Furthermore, please align your manuscript with our format requirements, which are summarized on the following checklist: <https://www.nature.com/documents/commsj-phys-style-formatting-checklist-article.pdf> Communications Earth & Environment formatting checklist

and also in our style and formatting guide <https://www.nature.com/documents/commsj-phys-style-formatting-guide-accept.pdf> Communications Earth & Environment formatting guide .

*** DATA: Communications Earth & Environment endorses the principles of the Enabling FAIR data project (<http://www.copdess.org/enabling-fair-data-project/>). We ask authors to make the data that support their conclusions available in permanent, publically accessible data repositories. (Please contact the editor if you are unable to make your data available).

All Communications Earth & Environment manuscripts must include a section titled "Data Availability" at the end of the Methods section or main text (if no Methods). More information on this policy, is available at <http://www.nature.com/authors/policies/data/data-availability-statements-data-citations.pdf>.

If a community resource is unavailable, data can be submitted to generalist repositories such as <https://figshare.com/> or <http://datadryad.org/> Dryad Digital Repository. Please provide a unique identifier for the data (for example a DOI or a permanent URL) in the data availability statement, if possible. If the repository does not provide identifiers, we encourage authors to supply the search terms that will return the data. For data that have been obtained from publically available sources, please provide a URL and the specific data product name in the data availability statement. Data with a DOI should be further cited in the methods reference section.

Please refer to our data policies at <http://www.nature.com/authors/policies/availability.html>

REVIEWER COMMENTS:

Reviewer #1 (Remarks to the Author):

see the PDF file in attachment

Reviewer #2 (Remarks to the Author):

Review of ms# COMMSENV-25-4373-T by Yamamoto et al.

Understanding the processes by which Earth's magnetic field reverses polarity has been a longstanding quest. This paper provides surprising new observational constraints: that polarity transitions can have durations of tens of thousands of years. The normal issues that make it difficult to reach such conclusions robustly are imprecise chronologies and magnetically poorly behaved samples within transition zones. These issues are well handled with a high-quality astronomically calibrated age model and magnetically well-behaved sediments, although I point to a few related matters below. Regardless, the potential for long polarity transition durations is well justified by presentation of new numerical geodynamo simulations. The results are clear and are well-presented by seasoned professionals, so there are no glaring deficiencies that need correction. I, therefore, recommend the paper for publication subject to minor revisions. The results provide a new view that will stimulate further debate and further studies. I have annotated the ms to help polish the writing. The annotated comments should be addressed carefully as well as the comments below.

1. When dealing with detailed records across thick stratigraphic intervals in IODP cores, I am always interested to see where core breaks and splice points lie in a composite stratigraphy. In this case, core breaks are irrelevant because the splice avoids them. Nevertheless, the record is plotted on a composite section — see my comments on Figure 2. This is all shown nicely in ED Fig. 5, but I often do not read a supplement and expect a main text to contain critical information. So, I would like to see labels added to Figure 2 to indicate the extent of the record from Hole A and that from Hole C with the interval of overlap indicated. Adding the requested labels would provide an appreciated level of transparency in data reporting in the main text.

2. Age model precision is critical when constraining event durations. While I find the age model convincing, uncertainties are important in science. Sedimentation rates will generally not be uniform between age tie points and there is always uncertainty in pinning tie points. I would like to see age uncertainties added to the polarity transition duration estimates. I know this is not straightforward, but our group has been doing Quaternary-style uncertainty estimation for the PETM, also with astronomical age models, so it is possible in this age range (see papers by Piedrahita et al.).

3. All of the evidence presented points to the palaeomagnetic data being of high quality. This is based on ChRM estimations from anchored PCA fits. I encourage the authors to read Heslop and Roberts (2016; JGR) about anchoring of PCA fits. In most cases, anchoring produces artificially low uncertainty compared to an unanchored fit, which cannot be justified statistically. If unanchored fits are used, MAD values increase. The vector component plots shown in ED Fig. 1 are presented honestly and have variable quality. Removing the anchoring constraint will make them look less good (which is typical of polarity transitions). I encourage the authors to consider this and its implications for their record. Heslop and Roberts have other papers on error propagation from the single sample level to VGP representations, which are also relevant here.

4. Thank you for showing the NRM-ARM slopes in ED Figs. 3 and 4. While the linear fits look reasonable, I can't help but notice that some of the slopes are curved. The FORC diagrams look like the sediments are dominated by magnetofossils, but it is hard to imagine that these sediments do not have additional terrigenous magnetic mineral contents (a double TV signal in low-temperature data would be indicative of such a component, for example). In sediments with significant contributions from both detrital and biogenic magnetite, we have found that it is useful to fit separate slopes to the low coercivity detrital component and to the higher coercivity biogenic component (Ouyang et al., 2014; Chen et al., 2017; both in G-cubed). Both tend to yield the same RPI results, but we have found that the biogenic component records the field substantially more efficiently than the detrital component, while Li and Yamazaki have published papers with the opposite result. I point this out for consideration. While the RPI estimation method used here is not a simple 'brute-force' normalization, further nuances deserve consideration. By the way, the method used looks like the "pseudo-Thellier" approach of Tauxe et al. (1995; GRL), which deserves citation.

5. I doubt if the statement on code availability (p. 30) will be acceptable. Open provision of code will be required.

6. Minor comments

- Please use U.K. rather than U.S. spellings for this journal (as indicated in my annotations).
- The claim on p. 4 that the reversal process occupies <1.3% of the reversal record did not make sense to me (see my failed calculation in the margin). If this number is important, please provide more of an explanation.
- Please ensure that the ED figures are called out in order in the main text and supplementary text.

d. Please check that each reference complies rigorously with journal style.

Andrew P. Roberts
October 19, 2025

** Visit Nature Portfolio's author and referees' website at www.nature.com/authors for information about policies, services and author benefits**

Communications Earth & Environment is committed to improving transparency in authorship. As part of our efforts in this direction, we are now requesting that all authors identified as 'corresponding author' create and link their Open Researcher and Contributor Identifier (ORCID) with their account on the Manuscript Tracking System prior to acceptance. ORCID helps the scientific community achieve unambiguous attribution of all scholarly contributions. You can create and link your ORCID from the home page of the Manuscript Tracking System by clicking on 'Modify my Springer Nature account' and following the instructions in the link below. Please also inform all co-authors that they can add their ORCIDs to their accounts and that they must do so prior to acceptance.

Version 1:

Decision Letter:

Dear Dr Yamamoto,

Your revised manuscript titled "Extraordinarily long duration of Eocene geomagnetic polarity reversals" has now been seen by our reviewers, whose comments appear below. In light of their advice we are delighted to say that we are happy, in principle, to publish a suitably revised version in Communications Earth & Environment.

We therefore invite you to revise your paper one last time to address the remaining concerns of our reviewers. At the same time we ask that you edit your manuscript to comply with our format requirements and to maximise the accessibility and therefore the impact of your work.

EDITORIAL REQUESTS:

*****Please take care to match our formatting and policy requirements. We will check revised manuscript and return manuscripts that do not comply. Such requests will lead to delays. *****

SUBMISSION INFORMATION:

OPEN ACCESS:

Communications Earth & Environment is a fully open access journal. Articles are made freely accessible on publication. For further information about article processing charges, open access funding, and advice and support from Nature Portfolio, please visit <https://www.nature.com/commsenv/open-access>

Link Redacted

Best regards,

Joao Duarte, PhD
Editorial Board Member
Communications Earth & Environment
orcid.org/0000-0001-7505-3690

Alireza Bahadori, PhD
Senior Editor
Communications Earth & Environment
Consulting Editor
Communications Sustainability

REVIEWERS' COMMENTS:

Reviewer #1 (Remarks to the Author):

I am very satisfied with the authors' correction and would recommend the manuscript for publication.

Reviewer #2 (Remarks to the Author):

Review of COMMSENV-25-4373A

I am at sea at present and quite busy, so my comments are brief. This is a revised paper that I have reviewed before. It is an interesting paper on a topic of wide interest that brings new observational constraints that supplement modelling results that help to understand the range of geomagnetic field behaviour to be expected when Earth's magnetic field undergoes reversal. All review comments have been addressed satisfactorily, and I am happy to recommend the paper for publication. I provide a lightly annotated scanned version of the manuscript with some minor tidying for the authors to do before they provide a final version to the publisher.

Andrew P. Roberts
NW Pacific Ocean
19/11/25

** Visit Nature Portfolio's author and referees' website at www.nature.com/authors for information about policies, services and author benefits**

Response to major and minor comments by the reviewer 1

1. Description of the morphology of the two polarity events. Whereas the authors endorse the paradigm of chaotic polarity reversals and have solid arguments for it, I see a contradiction in the author's attempt to classify the stages of the polarity reversals based on the simplified model by Valet et al. (2012) in terms of precursor, transit and rebound. Especially for the transition 2 presented in Figure 3 of the manuscript, the definition of the transit and rebound stages is somehow arbitrary and would deserve more caution. In spite of being fascinating by its simplicity, the study by Valet et al. (2012) was based on two strong assumptions (e.g. Glatzmaier & Coe, 2015). First, each polarity should have the same duration. Second, relying on the first assumption, each polarity reversal should display the same morphology. However, none of the individual record simultaneously showed a precursor and a rebound at the same time. If the authors' records for the transitions between chrons C18n.2n, C18n.1r and C18n.1n are inconsistent with the simplified model by Valet et al. (2012), they should not feel obliged to use Valet et al. (2012)'s model to describe their record. The assumptions made by Valet et al. (2012) may just be wrong.

We appreciate the comment. The simplified model proposed by Valet et al. (2012) in a Nature paper has had a strong influence on how palaeomagnetists interpret reversal processes, largely because no new high-resolution sedimentary records capturing reversals have been reported since its publication. When interpreting our Eocene reversal records, we adhered as closely as possible to this simplified model; for us, and many in community, the timing and synchronicity is less important than the directional structure of the reversal. However, as you pointed out, we observed some differing behaviors—such as multiple rebounds—that deviate from the model. To provide better context for these observations, we reorganized the fourth paragraph in the section “Middle Eocene palaeomagnetic polarity transition record” adding several sentences that incorporate your points, including the fact that the simplified model by Valet et al. (2012) is based on the two strong assumptions you described.

2. Time rescaling of the dynamo sequences. As mentioned by the authors, the analysed numerical dynamo simulations do not operate in the hypothetical force balance of the Earth's core (e.g. Schwaiger et al., 2019). With an Ekman number $E = \nu/\Omega D^2$ of 6.5×10^3 similar to the one used in the studies by Olson (2007) and Lhuillier et al. (2013), the employed dynamo simulations largely overestimate viscosity forces with respect to rotational forces. As a consequence, time rescaling is a delicate matter. Depending on the focus of the study, different timescales should be used for the rescaling of the time axis of the simulations: Alfvén

timescale if the purpose is to study wave propagation, magnetic diffusive timescale (or dipole-decay timescale) if the purpose is to study diffusive processes, advective timescale if the purpose is to study advective processes, etc. For the study of secular-variation processes thought to be controlled by the magnetic Reynolds number, the master constant $\tau_{SV} = \tau_\ell \times \ell$ of secular variation (where τ_ℓ is the correlation/reorganisation time for a spherical harmonic degree ℓ) is generally used (e.g. Christensen & Tilgner, 2004; Lhuillier et al., 2011). The simulations used by the authors operate with $R_m = 167\text{--}372$ whereas the value of R_m in the Earth's core is probably on the order of 1000. This means that a correction factor $R_m(\text{model})/R_m(\text{Earth})$ of 0,17–0,37 should be applied when rescaling the time using the dipole-decay time. It implies that the reversal durations presented in the manuscript may be overestimated by $R_m(\text{Earth})/R_m(\text{model})$, i.e. by a factor of 3–6. This caveat should be clearly mentioned in the manuscript. Alternatively, the authors may also claim that the reversal duration is controlled by diffusive processes as suggested by Lhuillier et al. (2013). If it is the case, such an assumption should be clearly stated in the manuscript.

Thank you very much for your very instructive comment. As you pointed out, time-rescaling from a dynamo run at a high viscosity is a very delicate matter. In rescaling time based on the dipole free decay time, we assume that reversal duration is controlled by diffusive processes as suggested in Lhuillier et al. (2013). We now clearly mention it in the first paragraph of the section "Variability of reversal durations from numerical geodynamo simulations" following the reviewer's comment. Also, some description about time-rescaling using the secular-variation timescale is added for additional instruction and clarity. Thank you for this recommendation.

Definition of the transitional times. It is always a debated matter in the palaeomagnetic community. The transitional times are sometimes detected using a fixed VGP cutoff 40° (e.g. Quidelleur et al., 1994) or of 45° (e.g. McFadden & Merrill, 1997). However, due to the latitudinal dependency of the VGP scatter, an adaptive VGP cutoff (e.g. Vandamme, 1994) is sometimes preferred. The choice of the authors to use a fixed VGP cutoff of 45° is perfectly fine. It is stated in the second part of the manuscript that it is a conservative way to define the transitional times. For clarity, please mention this caveat earlier in the manuscript (i.e. at lines 110–112).

We appreciate the comment. We have added the sentence of "It is noted that the cut-off angle of $\pm 45^\circ$ is a conservative criterion for defining magnetic polarity." in the third paragraph of the section "Middle Eocene palaeomagnetic polarity transition record".

Response to major and minor comments by the reviewer 2

1. When dealing with detailed records across thick stratigraphic intervals in IODP cores, I am always interested to see where core breaks and splice points lie in a composite stratigraphy. In this case, core breaks are irrelevant because the splice avoids them. Nevertheless, the record is plotted on a composite section — see my comments on Figure 2. This is all shown nicely in ED Fig. 5, but I often do not read a supplement and expect a main text to contain critical information. So, I would like to see labels added to Figure 2 to indicate the extent of the record from Hole A and that from Hole C with the interval of overlap indicated. Adding the requested labels would provide an appreciated level of transparency in data reporting in the main text.

We appreciate the comment. As explained in the subsection “Downcore VGP and RPI variations in the common depth scale” in METHODS section, the composite VGP and RPI variations were adopted from the 50.13-53.12 m CCFS_{Hole C} interval in Hole A and that of 53.12-58.13 m CCFS_{Hole C} in Hole C. Figure 2 has been redrawn to clearly indicate which intervals correspond to Hole A and Hole C.

2. Age model precision is critical when constraining event durations. While I find the age model convincing, uncertainties are important in science. Sedimentation rates will generally not be uniform between age tie points and there is always uncertainty in pinning tie points. I would like to see age uncertainties added to the polarity transition duration estimates. I know this is not straightforward, but our group has been doing Quaternary-style uncertainty estimation for the PETM, also with astronomical age models, so it is possible in this age range (see papers by Piedrahita et al.).

We appreciate the comment. We now provide uncertainties on duration estimates. We used the age-depth modeling routine, called “Undatable” (Lougheed and Obrochta, 2019), which has been successfully used for Quaternary series, and adapted here for our Eocene data. The advantage of the “Undatable” among other Quaternary age-depth modeling routines (e.g., Bchron, Clam, Bacon, etc) is that it accounts for uncertainties in both ages and depths, with higher precision even at shorter spatial and time scales. Uncertainty on our continuous depth sampling (discrete samples each 2 cm) that provides ultra high-resolution palaeomagnetic data, along with age errors inferred from variable sedimentation rates are included here in

the modeling to assess uncertainty on duration estimates. Results are summarized in Extended Data Fig. 6. These points are newly addressed in the 5th paragraph of the subsection "Middle Eocene palaeomagnetic polarity transition record". Thank you for this helpful suggestion.

3. All of the evidence presented points to the palaeomagnetic data being of high quality. This is based on ChRM estimations from anchored PCA fits. I encourage the authors to read Heslop and Roberts (2016; JGR) about anchoring of PCA fits. In most cases, anchoring produces artificially low uncertainty compared to an unanchored fit, which cannot be justified statistically. If unanchored fits are used, MAD values increase. The vector component plots shown in ED Fig. 1 are presented honestly and have variable quality. Removing the anchoring constraint will make them look less good (which is typical of polarity transitions). I encourage the authors to consider this and its implications for their record. Heslop and Roberts have other papers on error propagation from the single sample level to VGP representations, which are also relevant here.

We appreciate the comment. We agree that PCA with a fit anchored to the origin generally yields lower uncertainty compared to an unanchored fit. To evaluate the fidelity of the downcore VGP variations, we have newly included figures showing VGP variations with uncertainty proxies (MAD) for both anchored and unanchored fits (Extended Data Fig. 5a and 5b). In our case, the anchored fit indeed results in lower uncertainties; however, the overall characteristics of the VGP variation remain consistent between the two approaches. These points are newly addressed in the subsection "Downcore VGP and RPI variations in the common depth scale" of the METHODS section.

4. Thank you for showing the NRM-ARM slopes in ED Figs. 3 and 4. While the linear fits look reasonable, I can't help but notice that some of the slopes are curved. The FORC diagrams look like the sediments are dominated by magnetofossils, but it is hard to imagine that these sediments do not have additional terrigenous magnetic mineral contents (a double TV signal in low-temperature data would be indicative of such a component, for example). In sediments with significant contributions from both detrital and biogenic magnetite, we have found that it is useful to fit separate slopes to the low coercivity detrital component and to the higher coercivity biogenic component (Ouyang et al., 2014; Chen et al., 2017; both in G-cubed). Both tend to yield the same RPI results, but we have found that the biogenic component records the field substantially more efficiently than the detrital component, while Li and Yamazaki have published papers with the opposite result. I point this out for consideration. While the

RPI estimation method used here is not a simple 'brute-force' normalization, further nuances deserve consideration. By the way, the method used looks like the "pseudo-Thellier" approach of Tauxe et al. (1995; GRL), which deserves citation.

We appreciate the comment. Our method for estimating the relative paleointensity (RPI) from individual samples involves extracting the maximum possible linear segment from both NRM–ARM and NRM–IRM plots. Linearity is assessed using the correlation coefficient (r) of the linear regression, which in most cases exceeds 0.95. To define the linear segment, we typically use a coercivity range from 15–25 mT (lower bound) to 70–80 mT (upper bound). This selection is made to avoid the influence of secondary magnetization, which is commonly carried by grains with coercivities below 15–25 mT, most probably larger detrital magnetic grains. As a result, our RPI estimates are primarily derived from a coercivity interval of 20–80 mT, ensuring sufficient linearity and minimizing contamination from secondary components. These coercivities are within the biogenic soft and biogenic hard components for conventional magnetofossils and considering the findings of Ouyang et al. (2014) and Chen et al. (2017), we infer that the RPI signal in our samples predominantly reflects paleomagnetic records carried by the biogenic component. These points are newly addressed in the subsection "Determination of relative palaeointensity (RPI) of the geomagnetic field" of the METHODS section.

The method used in this study is based on the so-called "slope" method, which differs from the "pseudo-Thellier" approach proposed by Tauxe et al. (1995; GRL), where RPI is estimated using the relationship between NRM loss and ARM acquisition.

5. I doubt if the statement on code availability (p. 30) will be acceptable. Open provision of code will be required.

We appreciate the comment. The code used for the numerical geodynamo simulation is available via Zenodo (10.5281/zenodo.13917223), which will become publicly accessible upon publication. We updated the statement accordingly.

6. Minor comments

- a. Please use U.K. rather than U.S. spellings for this journal (as indicated in my annotations).*
- b. The claim on p. 4 that the reversal process occupies <1.3% of the reversal record did not make sense to me (see my failed calculation in the margin). If this number is important, please provide more of an explanation.*

c. Please ensure that the ED figures are called out in order in the main text and supplementary text.

d. Please check that each reference complies rigorously with journal style.

We appreciate the comment. We have addressed all the points you raised to the best extent possible.

Response to major and minor comments by the reviewer 1

I am very satisfied with the authors' correction and would recommend the manuscript for publication.

Thank you very much for your kind evaluation of our revisions in this second-round review. We would also like to express our sincere appreciation for your helpful and constructive comments in the first-round review, which greatly contributed to improving the quality of our manuscript. We are truly grateful for the time and effort you have devoted to reviewing our work.

Response to major and minor comments by the reviewer 2

All review comments have been addressed satisfactorily, and I am happy to recommend the paper for publication. I provide a lightly annotated scanned version of the manuscript with some minor tidying for the authors to do before they provide a final version to the publisher.

Thank you very much for your second round of review and for kindly recommending our paper for publication. We sincerely appreciate the annotated version of the manuscript you provided. We have carefully considered each of your comments and have incorporated the suggested minor revisions to the extent possible in preparing the revised manuscript.

Review of “Extraordinary long duration of Eocene geomagnetic polarity reversals” by Yuhji Yamamoto et al.

Characterising the morphology and duration of polarity reversals of the Earth’s magnetic field is essential to better constrain the working of the Earth’s dynamo. Historically, two contradictory paradigms have been proposed. On the one hand, influenced by the uniformitarian principle, palaeomagneticians have proposed that each polarity reversal should follow the same pattern, with the recurrent claim that each event should be preceded by a precursor and followed by a rebound (e.g. Valet et al., 2012). On the other hand, based on a better understanding of the chaotic nature of the geodynamo in the Earth’s outer core (e.g. Ryan & Sarson, 2008), geodynamicists followed by some palaeomagneticians alternatively proposed that polarity reversals should be complex events characterised by a stochastic path of the virtual geomagnetic poles (e.g. Jarboe et al., 2011) and reversal durations following a log-normal distribution (e.g. Lhuillier et al., 2013).

The submitted manuscript advances solid arguments in favour of the second paradigm. Based on the palaeomagnetic analysis of a deep-sea sedimentary core, the first part of the manuscript presents a high-resolution record of two polarity reversals during the Eocene. It constitutes in my opinion an important contribution to the field of palaeomagnetism and provides solid constraints to the modellers of the geodynamo. Based on numerical simulations, the second part proposes an analysis of the duration of polarity reversals for a set of dynamo models. This second part is in my opinion less convincing due to the overestimated viscosity in the employed numerical dynamo simulations, leading to an erroneous time rescaling from the dimensionless dipole-decay time to million years (see my point #2).

Major points

- 1) Description of the morphology of the two polarity events. Whereas the authors endorse the paradigm of chaotic polarity reversals and have solid arguments for it, I see a contradiction in the author’s attempt to classify the stages of the polarity reversals based on the simplified model by Valet et al. (2012) in terms of precursor, transit and rebound. Especially for the transition 2 presented in Figure 3 of the manuscript, the definition of the transit and rebound stages is somehow arbitrary and would deserve more caution. In spite of being fascinating by its simplicity, the study by Valet et al. (2012) was based on two strong assumptions (e.g. Glatzmaier & Coe, 2015). First, each polarity should have the same duration. Second, relying on the first assumption, each polarity reversal should display the same morphology. However, none of the individual records simultaneously showed a precursor and a rebound at the same time. If the authors’ records for the transitions between chrons C18n.2n, C18n.1r and C18n.1n are inconsistent with the simplified model by Valet et al. (2012), they

should not feel obliged to use Valet et al. (2012)'s model to describe their record. The assumptions made by Valet et al. (2012) may just be wrong.

2) Time rescaling of the dynamo sequences. As mentioned by the authors, the analysed numerical dynamo simulations do not operate in the hypothetical force balance of the Earth's core (e.g. Schwaiger et al., 2019). With an Ekman number $E = \nu/\Omega D^2$ of 6.5×10^3 similar to the one used in the studies by Olson (2007) and Lhuillier et al. (2013), the employed dynamo simulations largely overestimate viscosity forces with respect to rotational forces. As a consequence, time rescaling is a delicate matter. Depending on the focus of the study, different timescales should be used for the rescaling of the time axis of the simulations: Alfvén timescale if the purpose is to study wave propagation, magnetic diffusive timescale (or dipole-decay timescale) if the purpose is to study diffusive processes, advective timescale if the purpose is to study advective processes, etc. For the study of secular-variation processes thought to be controlled by the magnetic Reynolds number, the master constant $\tau_{SV} = \tau_\ell \times \ell$ of secular variation (where τ_ℓ is the correlation/reorganisation time for a spherical harmonic degree ℓ) is generally used (e.g. Christensen & Tilgner, 2004; Lhuillier et al., 2011). The simulations used by the authors operate with $Rm=167-372$ whereas the value of Rm in the Earth's core is probably on the order of 1000. This means that a correction factor $Rm(\text{model})/Rm(\text{Earth})$ of 0,17–0,37 should be applied when rescaling the time using the dipole-decay time. It implies that the reversal durations presented in the manuscript may be overestimated by $Rm(\text{Earth})/Rm(\text{model})$, i.e. by a factor of 3–6. This caveat should be clearly mentioned in the manuscript. Alternatively, the authors may also claim that the reversal duration is controlled by diffusive processes as suggested by Lhuillier et al. (2013). If it is the case, such an assumption should be clearly stated in the manuscript.

Minor point

Definition of the transitional times. It is always a debated matter in the palaeomagnetic community. The transitional times are sometimes detected using a fixed VGP cutoff of 40° (e.g. Quidelleur et al., 1994) or of 45° (e.g. McFadden & Merrill, 1997). However, due to the latitudinal dependency of the VGP scatter, an adaptive VGP cutoff (e.g. Vandamme, 1994) is sometimes preferred. The choice of the authors to use a fixed VGP cutoff of 45° is perfectly fine. It is stated in the second part of the manuscript that it is a conservative way to define the transitional times. For clarity, please mention this caveat earlier in the manuscript (i.e. at lines 110–112).

References:

- Christensen, U. R., & Tilgner, A. (2004). Power requirement of the geodynamo from ohmic losses in numerical and laboratory dynamos. *Nature*, *429*(6988), 169-171. 10.1038/nature02508
- Glatzmaier, G. A., & Coe, R. S. (2015). Magnetic Polarity Reversals in the Core. In P. L. Olson (Ed.), *Core Dynamics* (Vol. 8, pp. 279-295): Elsevier B.V.
- Jarboe, N. A., Coe, R. S., & Glen, J. M. G. (2011). Evidence from lava flows for complex polarity transitions: the new composite Steens Mountain reversal record. *Geophysical Journal International*, *186*, 580-602. 10.1111/j.1365-246X.2011.05086.x
- Lhuillier, F., Fournier, A., Hulot, G., & Aubert, J. (2011). The geomagnetic secular-variation timescale in observations and numerical dynamo models. *Geophysical Research Letters*, *38*(9), L09306. 10.1029/2011GL047356
- Lhuillier, F., Hulot, G., & Gallet, Y. (2013). Statistical properties of reversals and chrons in numerical dynamos and implications for the geodynamo. *Physics of the Earth and Planetary Interiors*, *220*, 19-36. 10.1016/j.pepi.2013.04.005
- McFadden, P. L., & Merrill, R. T. (1997). Asymmetry in the reversal rate before and after the Cretaceous Normal Polarity Superchron. *Earth and Planetary Science Letters*, *149*, 43-47. 10.1016/S0012-821X(97)00061-7
- Olson, P. L. (2007). Gravitational dynamos and the low-frequency geomagnetic secular variation. *Proceedings of the National Academy of Sciences of the United States of America*, *104*(51), 20160-20166. 10.1073/pnas.0709081104
- Quidelleur, X., Valet, J.-P., Courtillot, V., & Hulot, G. (1994). Long-term geometry of the geomagnetic field for the last five million years: An updated secular variation database. *Geophysical Research Letters*, *21*(15), 1639-1642. 10.1029/94GL01105
- Ryan, D. A., & Sarson, G. R. (2008). The geodynamo as a low-dimensional deterministic system at the edge of chaos. *European Physical Letters*, *83*(4). 10.1209/0295-5075/83/49001
- Schwaiger, T., Gastine, T., & Aubert, J. (2019). Force balance in numerical geodynamo simulations: a systematic study. *Geophysical Journal International*, *219*, 101-114. 10.1093/gji/ggz192
- Valet, J.-P., Fournier, A., Courtillot, V., & Herrero-Bervera, E. (2012). Dynamical similarity of geomagnetic field reversals. *Nature*, *490*(7418), 89-93. 10.1038/nature11491
- Vandamme, D. (1994). A new method to determine paleosecular variation. *Physics of the Earth and Planetary Interiors*, *85*(1-2), 131-142. 10.1016/0031-9201(94)90012-4

**Extraordinarily long duration of Eocene geomagnetic polarity**
**reversals**

Yuhji YAMAMOTO *¹

Slah BOULILA^{2,3}

Futoshi TAKAHASHI⁴

Peter C. LIPPERT⁵

1: Marine Core Research Institute, Kochi University, Kochi 783-8502, Japan

2: Sorbonne Université, CNRS, Institut des Sciences de la Terre Paris, ISTeP, F-75005

Paris, France.

3: ASD, IMCCE-CNRS UMR 8028, Observatoire de Paris, UPMC, 77 avenue

Denfert-Rochereau, 75014 Paris, France.

4: Department of Earth and Planetary Sciences, Kyushu University, Fukuoka

819-0395, Japan

5: Department of Geology & Geophysics, University of Utah, Utah, 84112, USA

**Abstract:**

One of the most intriguing attributes of Earth's magnetic field is that it reverses polarity.
Sedimentary palaeomagnetic records for the last four reversals back to 1.78 million
21 years ago and volcanic palaeomagnetic records for seven field reversals mainly from
22 0.78-16.7 million years ago indicate that the reversal process typically occurs over ten
thousand or fewer years and follows a common pattern. However, it remains unclear if
the deep-time behaviour of geomagnetic reversals is similar. We present an
exceptionally resolved deep sea sedimentary palaeomagnetic record that captures two
polarity transitions that occurred ~40 million years ago. These polarity transitions occur
over 18 and 70 thousand years and, thus, are significantly longer than the typical 10 kyr
duration for younger reversals. Longer-duration transitions like these are calculated in
numerical geodynamo models. Our findings suggest that variable reversal durations are
an intrinsic property of the geodynamo. This variability predicts ^{that} polarity transition
duration, ^{can be} ~~are~~ much longer than 10 kyr. Occasional prolonged intervals of transitional,
and weaker, geomagnetic fields would have exposed Earth's Eocene environments to
greater high-energy radiation from the sun for longer intervals of time, with potential
impacts on biota.

a rebound phase lasting ~ 2.5 kyr⁹. These records also indicate that the total reversal
duration in direction is ~ 9 kyr⁹. The most detailed sedimentary record of the last
geomagnetic reversal (Brunhes-Matuyama Transition, ~ 0.77 Ma) dates the reversal
duration in direction to ~ 11 kyr¹⁰. Thus, it is commonly assumed that geomagnetic
reversals in direction last ~ 10 kyr from start to finish based on what we observe and
measure from reported palaeomagnetic records.

Approximately 540 geomagnetic reversals have occurred over the last ~ 170 million
years¹¹. The estimated ~ 10 kyr duration of the entire reversal process in direction
deduced from just seven reversal records^{8,9}, ~~that is~~ ^{which} ~~only~~ ^{represent} ~~based on~~ $< 1.3\%$ of ~~the~~

reversals ~~occurred~~ over the last ~ 170 million years. Thus, this duration estimation may
not represent intrinsic geomagnetic behaviour. For example, some numerical
geodynamo simulations (without data assimilation) suggest that reversal durations
follow a log-normal law, thus permitting more temporal variability and longer reversal
durations than commonly assumed¹². Although geodynamo simulations do not perfectly
mimic Earth-like dynamos, they nonetheless provide time-series data that are
remarkably similar in behaviour to the Brunhes-Matuyama Transition¹³. Here we
present a highly resolved sedimentary record of two Eocene reversals that occurred ~ 40

OK - I get
if new.

p. 3 not scanned - no edits

4

million years ago, which have extraordinarily long duration of polarity reversals in

~~direction: the transition into polarity Chron C18n.1r lasted 18±3 kyr and the transition~~

~~out of Chron C18n.1r persisted for 70±6 kyr.~~ Although numerous sedimentary

palaeomagnetic records have been reported, high-resolution sedimentary records like

those reported here are rare. Previously, the oldest similarly high-resolution sedimentary

records of geomagnetic reversals were dated to ~2.58 Ma¹⁴ and ~3.33 Ma^{15,16}, giving

directional reversal duration^s of ~10 kyr. We compare our observations of Eocene data to

models from numerical geodynamo simulations to further explore directional reversal

duration variability.

Middle Eocene palaeomagnetic polarity transition record

Integrated Ocean Drilling Program (IODP) Expedition 342 recovered Palaeogene

sedimentary drift sequences from the Newfoundland ridges in the northwest Atlantic

85 ^ocean¹⁷. The studied Site U1408 sediments were deposited at average rates of ~2.4

86 cm/kyr and have prominent lithological alternations that can be detected robustly from

87 multiple chemical and physical proxy data (Fig. 1)¹⁸. We used scanning

X-ray-fluorescence-measured Ca/Fe ratios to calibrate the geochronology of most of the

middle Eocene sediments, based on the stable 173-kyr-obliquity cycle of Earth's axial

The introduction is not the place to tell your results. It is where you convince readers of the value of what you are doing.

5

[revised manuscript text omitted]

ideally you want to smooth
noise not signal.

**Determination of virtual geomagnetic pole (VGP) latitude**

Stepwise AF demagnetization results of NRM were used to calculate characteristic
remanent magnetization (ChRM) directions of the samples using principal component
analysis (PCA) with a fit anchored to the origin⁴⁸. Except for a few samples, ChRM fits
have maximum angular deviation (MAD) values less than 10 ° (Extended Data Fig. 1).
The cores were recovered from the seafloor using the FlexIt core orientation tool, which
provides the magnetic tool face (MTF) orientation value giving the angle between
geomagnetic north and the double line on the core liner for each core⁴⁹. Thus, the

p. 26 not scanned

481 variations are plotted on the between-hole composite depth scale in meters² CCFS,
482 cm-scale offsets are evident between Hole A and Hole C for the overlapping interval
(~52.2-54.8 m CCFS). Calibration of the geochronology was mainly made using the
X-Ray fluorescence-derived Ca/Fe record from Hole C¹⁸. We held the Hole C depths
fixed ($CCFS_{\text{Hole C}}$) and shifted the Hole A depth ($CCFS_{\text{Hole A}}$) with respect to the Hole C
depth to match the variation between Hole A and Hole C. We found that a 4-cm
upward-shift ($CCFS_{\text{Hole A}} - 0.04$ m) produces the best match for variations between the
two holes, so the Hole A depth was correlated to the Hole C depth ($CCFS_{\text{Hole C}}$) in this
way (Extended Data Fig. 5). RPI values from Hole C were rescaled to have the same
average as those of Hole A for the overlapping interval. The composite VGP and RPI
variations were adopted from the 50.13-53.12 m $CCFS_{\text{Hole C}}$ interval in Hole A and that
of 53.12-58.13 m $CCFS_{\text{Hole C}}$ in Hole C.

The VGP variation was interpreted based on palaeomagnetic directions determined
using PCA with a fit anchored to the origin. This approach generally yields lower
uncertainty compared to an unanchored fit⁵². In our case, the anchored fit indeed
resulted in lower uncertainties than the unanchored fit; however, the overall
characteristics of the VGP variation remain consistent between the two methods

pp. 28-29 not scanned

**Extended data figure/table legends**

**Extended Data Fig. 1**
Representative orthogonal vector component plots of AF demagnetization results for the
NRM of the studied samples. Solid (open) circles indicate horizontal (vertical)
projections. Grey circles represent data not used for PCA. Black dashed lines are
best-fits to the data.

**Extended Data Fig. 2**
Representative results of low-temperature magnetometry on sample fractions selected
from three horizons (a-c) and FORC diagrams for the same horizons (d-f).

**Extended Data Fig. 3**
Representative NRM-ARM diagrams for AF demagnetization results for the studied
samples. Best-fit slopes are determined and their values with correlation coefficient (r)
and number of data points (N) are indicated.

**Extended Data Fig. 4**
Representative NRM-IRM diagrams for AF demagnetization results from the studied

samples. Best-fit slopes are determined and their values with correlation coefficient (r)
and number of data points (N) are indicated.

**Extended Data Fig. 5**

Downcore variations of VGP³ (a, b) and RPI (c) plotted on the common depth scale. The
between-hole composite depth scale in meters² CCSF was further adjusted to have the
best matches in variations between Hole A (blue) and Hole C (red). Hole A depths were
correlated to Hole C depths ($CCFS_{Hole\ C}$) by shifting the Hole A record 4 cm upward
($CCFS_{Hole\ A} - 0.04\ m$). VGP variations are shown with uncertainty proxies (MAD) based
on PCA, with and without anchoring the fit to the origin, in panels 'a' and 'b',
respectively.

**Extended Data Fig. 6**

Age-depth model of the studied Eocene stratigraphic interval using the "Undatable"
age-depth modeling routine^{27b}. The eight selected age-depth points (indicated by purple
stars) are from younger to older ages: top of the studied stratigraphic interval, base of
C18n.1n, base of transition 2, base-minimum of La2011 s3-s6 astronomical cycle No.
1229¹⁸, base of C18n.1r, base of transition 1, base-minimum of La2011 s3-s6

astronomical cycle No. 1230¹⁸, and base of the studied stratigraphic interval. (a) using
eight age-depth points. (b) using six age-depth points (i.e., without the top and base of
the stratigraphic interval). Uncertainties are provided with 1-sigma.

**Extended Data Fig. 7**

Typical examples of time-series for (a) simple and (b) complex polarity reversals in a
numerical dynamo model (case-R2). Tilt angle represents dipole deviations from the
rotation axis. Denoted in green is the duration of polarity transition according to the
criteria adopted in this study.

**Extended Data Table. 1**

Results of numerical geodynamo modeling.

Run ID: represents each dynamo case; *Ek*: Ekman number; *Ra*: Rayleigh number; Total
run time: simulation run time in the unit of dipole free decay time; Mean duration: mean
of the sampled reversal duration; Range of durations: minimum and maximum values of
the reversal duration; *Rm*: Magnetic Reynolds number in terms of volume-averaged
velocity; *A*: Elsasser number in terms of volume-averaged magnetic field.

This information should go below the Table.

Extraordinary ^{il} long duration of Eocene geomagnetic polarity reversals] ^{bold}

Yuhji YAMAMOTO * ¹

Slah BOULILA ^{2,3}

Futoshi TAKAHASHI ⁴

Peter C. LIPPERT ⁵

1: Marine Core Research Institute, Kochi University, Kochi 783-8502, Japan

2: Sorbonne Université, CNRS, Institut des Sciences de la Terre Paris, ISTeP, F-75005

Paris, France.

3: ASD, IMCCE-CNRS UMR 8028, Observatoire de Paris, UPMC, 77 avenue

Denfert-Rochereau, 75014 Paris, France.

4: Department of Earth and Planetary Sciences, Kyushu University, Fukuoka

819-0395, Japan

5: Department of Geology & Geophysics, University of Utah, Utah, 84112, USA

Need to use U.K. spellings throughout

**Abstract:**

One of the most intriguing attributes of Earth's magnetic field is that it reverses polarity.

Sedimentary paleomagnetic records ^{for} the last four reversals back to 1.78 million years

ago and volcanic paleomagnetic records ^{for} seven field reversals mainly from 0.78-16.7

million years ago ^{indicate} ~~show~~ that the reversal process typically occurs over ten thousand or

fewer years and follows a common pattern ~~in direction~~. However, it remains unclear if

the deep-time behavior ^{of} geomagnetic reversals is similar. We present an exceptionally

resolved deep sea sedimentary paleomagnetic record that captures two polarity

transitions that occurred ~40 million years ago. These polarity transitions occur over 18

and 70 thousand years and ^{for younger reversals} thus are significantly longer than the typical 10 kyr duration.

Longer-duration transitions like these are calculated in numerical geodynamo models.

Our findings suggest that variable reversal durations are an intrinsic property of the

geodynamo ~~and this~~ ^{duration are} variability predicts polarity transitions ~~that occur~~ much longer than

10 kyr. Occasional prolonged intervals of transitional, and weaker, geomagnetic fields

would have exposed ^{Eocene} Earth's environments to greater high-energy radiation from

the sun for longer intervals of time, with potential ^{to} impacts ~~on biota~~.

Main text ✓

~~_____~~ Delete

Earth's ~~geo~~ magnetic field is one of the most intrinsic properties of ^{our} ~~the~~ planet. The field

is generated by fluid motion of an iron-nickel outer core and may have started ~4.2

billion years ago¹. The geomagnetic field changes its polarity stochastically and may

have begun reversing ~3.5 billion years ago². Dipole energy is drastically reduced

during geomagnetic reversals^{3,4}, resulting in dynamic variations in both the ^{field} intensity

and direction ~~of the field~~. Whereas volcanic records can provide precise, but temporally

incomplete records of direction and intensity⁵, marine sediments can capture continuous

variations ~~of the~~ geomagnetic field⁶ and provide complementary and fundamental

information of the reversal process⁷. Sediment records ^{for} ~~of~~ the last four reversals (back to

1.78 million years ago (Ma)) suggest that the entire reversal process in direction (stable

polarities of opposite directions) takes approximately 1-12 thousand years (kyr)

depending on the latitude at which the reversal is observed⁸. Semi-continuous

[↑] paleomagnetic records of seven field reversals obtained from sequential lava flows on

land for two intervals from 0.78-16.7 Ma and at 180 Ma suggest that the reversal

process comprises three successive and distinct phases of magnetic direction and dipole

intensity: a precursor phase lasting ~2.5 kyr; a main transition phase lasting ~1 kyr; and

a rebound phase lasting ~ 2.5 kyr⁹. These records also indicate that the total duration of a
reversal in direction is ~ 9 kyr⁹. The most detailed sedimentary record of the last
geomagnetic reversal (Brunhes-Matuyama Transition, ~ 0.77 Ma) dates the reversal
duration in direction to ~ 11 kyr¹⁰. Thus, it is commonly assumed that geomagnetic
reversals in direction last ~ 10 kyr from start to finish based on what we observe and
measure from the paleomagnetic records reported so far.

Approximately 540 geomagnetic reversals have occurred over the last ~ 170 million
years¹¹. Thus, the estimated ~ 10 kyr duration of the entire reversal process in direction
occupies $< 1.3\%$ of the reversal record^{8,9} and may not represent intrinsic

$$\frac{540 \times 10 \text{ kyr}}{170 \text{ Ma}} \approx 3.2\%$$

geomagnetic behavior. For example, some numerical geodynamo simulations (without
data assimilation) suggest that reversal durations follow a log-normal law, thus
permitting more temporal variability and longer reversal durations than commonly
assumed¹². Although geodynamo simulations do not perfectly mimic Earth-like
dynamos, they nonetheless provide time-series data that are remarkably similar in
behavior to the Brunhes-Matuyama Transition¹³. Here we present a highly resolved
sedimentary record of two Eocene reversals that occurred ~ 40 million years ago, which have
showing extraordinary long duration of polarity reversals in direction: the transition into

well-represented part of the spectrum of behaviour

polarity Chron C18n.1r lasted ~18 kyr and the transition out of Chron C18n.1r persisted
for ~70 kyr. Although numerous sedimentary paleomagnetic records have been reported,
high-resolution sedimentary records like those reported here are rare. Previously, the
oldest similarly high-resolution sedimentary records of geomagnetic reversals were
dated to ~2.58 Ma¹⁴ and ~3.33 Ma^{15,16}, giving ~~duration of the reversals~~ ^{directional} ~~in direction of~~
~10 kyr. We compare our observations of Eocene Data to models from numerical
geodynamo simulations to further explore ~~the variability~~ ^{directional} reversal durations ~~in~~ ^{directional}
~~direction~~

**Middle Eocene paleomagnetic polarity transition record**

Integrated Ocean Drilling Program (IODP) Expedition 342 recovered Paleogene
sedimentary drift sequences from the Newfoundland ridges in the northwest Atlantic¹⁷.

The studied Site U1408 sediments were deposited at average rates of ~2.4 cm/kyr and
~~show~~ ^{have} prominent lithological alternations ~~which~~ ^{that} can be ~~robustly~~ ^{robustly} detected ~~by~~ ^{from} multiple
chemical and physical proxy data (Fig. 1)¹⁸. We used scanning
X-ray-fluorescence-measured Ca/Fe ratios to calibrate the geochronology of most of the
middle Eocene sediments, based on the stable 173-kyr-obliquity cycle of Earth's axial

tilt amplitude modulation. The robust geochronology helps to elucidate the variation of the
relative paleointensity (RPI) of the geomagnetic field during the middle Eocene
(between ~38.4 and ~43.2 Ma), although the signal is smoothed because measurements
were made on continuous sample sections¹⁹. We reanalyzed the ~~reversed~~ ^{polarity transition} intervals at the
beginning and end of Chron C18n.1r using discrete samples collected approximately
each 2 cm over 8 m of section from Holes A and C at Site U1408. Results from discrete
samples avoid smoothing artifacts in the time series of direction and intensity change
and allow for more, and more precise, paleo- and rock magnetic measurements.

~~The~~ ^{the} measurements of the natural remanent magnetization (NRM) revealed a characteristic
remanent magnetization (ChRM) component after removal of ~~the~~ ^a secondary remanent
magnetization typically by 20 mT (Extended Data Fig 1). Our rock magnetic
measurements indicate that the remanent magnetizations are carried mainly by primary
biogenic magnetite, as evidenced by Verwey transitions at ~100 K²⁰ and distinctive
central-ridges in first-order reversal curve (FORC) diagrams²¹ (Extended Data Fig 2).
Thus, the ChRM components represent primary paleomagnetic directions from which
we calculated the latitudes of the positions of the north virtual geomagnetic pole (VGP).
We used measurements of ~~the~~ ^{the} anhysteretic remanent magnetization (ARM) and isothermal

More important papers in this regard are Egli et al. (2010; G-cubed) and Roberts et al. (2012; JGR). You have room to add citations in this journal.

Reported earlier by Chong et al. (2016; JGR) - 1
Suggest you cite both papers because the latter paper validates these results

Magnetic extraction & TEM observations are still considered the best evidence

remanent magnetization (IRM) to calculate ratios of NRM to ARM (Extended Data Fig
3) and NRM to IRM to evaluate RPI (Extended Data Fig 4).

*relative paleointensity* / I don't see a definition for this above

Downcore VGP and *(RPI)* variations from Holes A and C were calculated and plotted *as*
the common depth scale for Site U1408 (Extended Data Fig 5). ~~an~~ interval *showing* the

VGP latitude higher (lower) than ~~45~~ *45° (-45°)* degree with continuity was then assigned *as a*

normal (reversed) polarity interval; other intervals ~~were~~ *are* interpreted to record polarity

transitions. ~~Figure 2 shows~~ *↑* composite VGP and RPI record *is shown in figure 2* with ~~the~~ interpreted

geomagnetic polarity chrons assigned based on the biostratigraphy and
magnetostratigraphy of Site U1408^{22,23}. Two polarity transition intervals are identified:

the end of normal-polarity subchron C18n.2n to the onset of reversed-polarity subchron

C18n.1r occurs over approximately *a interval* 50 cm (transition 1; Fig. 2b), and the end of C18n.1r

to the onset of normal-polarity subchron C18n.1n occurs over an interval of 170 cm
(transition 2; Fig. 2b). These transitions are associated with low RPI (Fig. 2c), which is

typical of polarity transitions observed *over* ~~from~~ the last ~ 50 million years^{19, 24-26}.

The geochronology based on the 173-kyr obliquity amplitude modulation cycle¹⁸

provided calibrated ages of 39.58 *9* - 39.60 *1* Ma for transition 1 and 39.43 *1* - 39.50 *1* Ma

for transition 2 (Fig. 3a). Thus, deposition rates are estimated to ^{be} 2.4-2.8 cm/kyr for these
intervals. For transition 1, we recognize only a main transition that lasts for ^{ed} ~2 kyr ^{with}
two rebound phases that last ^{ed} for ~11 and ~5 kyr (Fig. 3b). RPI remains low for ~38 kyr
(Fig. 3c). The occurrence of ~~the~~ multiple rebounds is not unprecedented: this behavior ^{is}
also reported for the Bruhnes-Matayama reversal²⁷, and we suggest that it may ^{occur} more
common^{ly}. All three reversal phases⁹ are recognized in transition 2 (Fig. 3b): the
precursor phase lasts ^{ed} ~22 kyr, the main transition occurs ^{ed} over ~9 kyr, and the rebound
phase comprises three intervals ^{with} ^{ed} ~14 kyr, ~13 kyr, and ~16 kyr ^{ed} RPI remains low for
~70 kyr (Fig. 3c). In summary, the duration of the entire reversal process in direction ^{is}
estimated to be ~18 kyr for transition 1 and ~70 kyr for transition 2. These are
significantly longer than ^{the} typically ~~assumed~~ duration ^{of} ~10 kyr ^{ed} based on published
paleomagnetic records.

Variability of reversal durations from numerical geodynamo simulations

The typical ~~assumed~~ duration of ~10 kyr for ~~the~~ ^{directional} geomagnetic reversals ~~in direction~~ is
derived from a composite record of 7 geomagnetic reversals^{8,9}, or only ~1 % of the
reversals that have occurred ^{over} ~~during~~ the last ~180 million years. Numerical geodynamo

simulations suggest that reversal durations are distributed log-normally¹², such that
longer transition durations are possible and expected even if they are not yet observed.
We evaluate this with our own numerical geodynamo model, which does not use data
assimilation. Our model produced 160 reversal events. We define reversal durations in
direction by the interval ~~when~~ ^{during which} the dipole axis stays within ~~the~~ ^{the} $\pm 45^\circ$ ~~latitude~~ ^{latitude}^{28,29}. This is a
conservative definition that yields shorter reversal durations. ~~Results~~ ^{Results} (Extended Data
Table 1) ~~show that~~ ^{show that} reversal durations in direction vary among runs with an average from
0.18 to 0.64 in the non-dimensional unit (Fig. 4). The upper bounds of these durations
are between 0.82 and 3.30, ~~suggesting~~ ^{which} that the longest reversal duration ranges from
~33 to 130 kyr when we use ~~the~~ ^a dipole free decay time of ~40 kyr to rescale the
non-dimensional time unit. We note that ~~the~~ transition durations would even be longer if
we used ^a higher electrical conductivity ~~of the~~ ^{value for} Earth's core³⁰, or a less conservative
~~definition of~~ ^{definition of} reversal.
The ~~results~~ ^{estimated} ~18 and ~70 kyr duration ^{respectively} for transition 1 and 2, are both admissible in
view of these numerical dynamo models. ^{results} We cannot exclude uncertainty, bias, or both
in calibrating time from dimensional units due to the vast differences in parameters
between ~~the~~ Earth's core and numerical dynamo models³¹. Results from numerical

dynamos suggest that reversals with long durations are characterized by complex
behaviors of the dipole axis, similar to our observations of field behavior during
transition 2, in which repeated multiple precursor, transitional, and rebound phases
occur during a single field reversal (Extended Data Fig 6). Although such behaviors are
qualitatively similar among the reversals¹³, mechanisms for such complex dipole
variations remain unclear and we cannot yet deduce any physics-based constraints on
typical timescales for ^{directional} reversal durations ~~in direction~~. More realistic and advanced
dynamo models with relevant force balance³² are necessary to explore what controls the
duration ~~of a reversal~~. However, it is still computationally impractical to use ~~the~~
state-of-the-art dynamo models to obtain a sufficient number of reversals to assess their
statistical properties, including reversal durations³³. Thus, our ~~current~~ ^{of} approach using a
modest parameter ~~for~~ viscosity [↓] is one ~~of the~~ reasonable ways [↓] to explore reversal
duration. Effects of core-mantle boundary heat flux variation and/or a stable stratified
layer at the top of the core, which are not ~~taken into~~ ^{ed} consideration in the present models,
may play a role in controlling reversal durations [↓], ~~as well as~~ reversal frequency and
geomagnetic secular variation^{28,34}.

**Implications**

~~Our findings suggest that variable reversal durations are an intrinsic property of the geodynamo. Moreover, this variability predicts polarity transitions that occur much longer than 10 kyr. The middle Eocene paleomagnetic polarity transition record we obtained from Site U1408 is presently at ~41°N; its paleolatitude at the time of sediment deposition (assuming a time-averaged geocentric axial dipole (GAD) field structure) is estimated to be ~22°N (measured inclination of ~40°). If the latitude dependence of reversal durations inferred from the last four reversals, namely, greater durations observed at higher latitudes, is also an intrinsic long-term property of the geodynamo, then reversal durations are expected to be even longer than ~70 kyr at mid to high latitude regions for transition 2. Some research suggests that the Eocene geomagnetic field was characterized by non-GAD field components larger than the recent field, particularly larger octupole contributions or by a more non-axial dipole. Deviations from a GAD field are expected to underestimate the true latitude of reversal observation (the sampling site) and produce more non-typical reversal behavior. Therefore, our results may be explained in part by non-GAD field contributions. Regardless, we observe two reversals that have the same directional structure as recent reversals and with durations that far exceed previous chronologies. Occasional prolonged intervals of~~

Delete

use a proper degree symbol

Our findings suggest that variable reversal durations are an intrinsic property of the

geodynamo. Moreover, this variability predicts polarity transitions that occur much

can be

longer than 10 kyr. ~~The middle Eocene paleomagnetic polarity transition record we~~

~~obtained from~~ Site U1408 is presently at ~41°N; its paleolatitude at the time of sediment

deposition (assuming a time-averaged geocentric axial dipole (GAD) field structure) is

estimated to be ~22°N (measured inclination of ~40°). If the latitude dependence of

reversal durations inferred from the last four reversals, namely, greater durations

observed at higher latitudes, is also an intrinsic long-term property of the geodynamo,

then reversal durations are expected to be even longer than ~70 kyr at mid to high

latitude regions for transition 2. Some research suggests that the Eocene geomagnetic

field was characterized by non-GAD field components larger than the recent field,

particularly larger octupole contributions or by a more non-axial dipole. Deviations

from a GAD field are expected to underestimate the true latitude of reversal observation

(the sampling site) and produce more non-typical reversal behavior. Therefore, our

results may be explained in part by non-GAD field contributions. Regardless, we

observe two reversals that have the same directional structure as recent reversals and

with durations that far exceed previous chronologies. Occasional prolonged intervals of

196 transitional, and weaker, geomagnetic fields, with or without large non-GAD
contributions, would have exposed Eocene environments to greater high-energy particle
radiation from the sun for longer intervals of time such ~~like~~^{as} in the Ediacaran^{37,38}. This
prolonged exposure may have influenced geochemical cycles, surface processes, and
biota of the Eocene ~~and~~^{and} testing such teleconnections merits further attention.

Delete blank
space

Please check journal reference format & ensure rigorously that all references comply.

**Main references**

- 1. J.A. Tarduno, R.D. Cottrell, R.K. Bono, H. Oda, W.J. Davis, M. Fayek, O.V. Erve, F.
Nimmo, W. Huang, E.R. Thern, S. Fearn, G. Mitra, A.V. Smirnov, E.G. Blackman,
Paleomagnetism indicates that primary magnetite in zircon records a strong
Hadean geodynamo. Proc. Natl. Acad. Sci. USA, 117, 2309–2318 (2020).
- 2. A.J. Biggin, M.J. de Wit, C.G. Langereis, T.E. Zegers, S. Voûte, M.J. Dekkers, K.
Drost, Palaeomagnetism of Archaean rocks of the Onverwacht Group, Barberton
Greenstone Belt (southern Africa): ^eEvidence for a stable and potentially reversing
geomagnetic field at ca. 3.5 Ga. Earth Planet. Sci. Lett. 302, 314–328 (2011).
- 3. P. Dagley, E. Lawley, ^aPaleomagnetic evidence for the transitional behaviour of the
geomagnetic field. Geophysical Journal of the Royal Astronomical Society 36,
213 ^{use an} 577–598 (1974). ^{m-dash} for consistency
- 4. H. Amit, R. Leonhardt, J. Wicht, Polarity ^RReversals from ^PPaleomagnetic
215 ^oObservations and ⁿNumerical ^dDynamo ^sSimulations. Space Sci. Rev. 155, 293–335
(2010).
- 5. R.S. Coe, J.M.G. Glen, The complexity of reversals, in Timescales of the
Paleomagnetic Field: American Geophysical Monograph 145, edited by J. E. T.
Channell, D. V. Kent, W. Lowrie and J. Meert, pp. ^o12, American Geophysical

- Union, Washington, D.C. (2004)
- 6. D.V. Kent, Post-depositional remanent magnetization in deep sea sediment, Nature
246, 32–34 (1973)
- 7. C. Laj, A. Mazaud, R. Weeks, M. Fuller, E. Herrero-Bervera, Geomagnetic reversal
paths. Nature 447, 351 (1991).
- 8. B.M. Clement, Dependence of the duration of geomagnetic polarity reversals on site
latitude. Nature, 428, 637-640 (2004). *an dash*
- 9. J.P. Valet, A. Fournier, V. Courtillot, E. Herrero-Bervera, Dynamical similarity of
geomagnetic field reversals. Nature 490, 89-94 (2012).
- 10. Y. Haneda, M. Okada, Y. Suganuma, T. Kitamura, A full sequence of the Matuyama–
Brunhes geomagnetic reversal in the Chiba composite section, Central Japan.
Progress in Earth and Planetary Science 7, 44 (2020).
- 11. J.G. Ogg, Geomagnetic Polarity Time Scale, In Geologic Time Scale 2020 edited by
Gradstein, F.M., Ogg, J.G., Schmitz, M.D., Ogg, G.M., Elsevier 1, 159-192 (2020)
- 12. F. Lhuillier, G. Hulot, Y. Gallet, Statistical properties of reversals and chrons in
numerical dynamos and implications for the geodynamo. *Physics of the Earth and
Planetary Interiors*, 220, 19–36 (2013).
- 13. P.L. Olson, G.A. Glatzmaier, R.S. Coe, Complex polarity reversals in a geodynamo

- model. ~~Earth and Planetary Science Letters~~ 304, 168-179 (2011).
- 14. M. Ohno, F. Murakami, F. Komatsu, Y. Guyodo, G. Acton, T. Kanamatsu, H.F.
Evans, F. Nanayama, Paleomagnetic directions of the Gauss-Matuyama polarity
transition recorded in drift sediments (IODP Site U1314) in the North Atlantic.
Earth Planets Space 60, e13-e16 (2008).
- 15. Y. Haneda, M. Okada, A record of the lower Mammoth geomagnetic polarity
reversal from a marine succession in the Boso Peninsula, central Japan.
~~Geophysical Journal International~~ 228, 461-476 (2022).
- 16. A. Tanimoto, M. Okada, R. Hayashi, Reconstruction of high-resolution
paleomagnetic variations in the middle Gauss chronozone, including the upper and
lower boundaries of the Mammoth reversed subchronozone. Earth, Planets ~~and~~
Space 76, 171 (2024).
- 17. Expedition 342 Scientists. Paleogene Newfoundland sediment drifts. IODP Prel.
Rept. 342, doi:10.2204/iodp.pr.342.2012 (2012).
- 18. S. Boulila, M. Vahlenkamp, D. De Vleeschouwer, J. Laskar, Y. Yamamoto, H. Palike,
S.K. Turner, P.F. Sexton, T. Westerhold, U. Rohl, Towards a robust and consistent
middle Eocene astronomical timescale. Earth Planet. Sci. Lett. 486, 94-107 (2018).
- 19. Y. Yamamoto, H. Fukami, P.C. Lippert, Eocene relative paleointensity of the

- geomagnetic field from Integrated Ocean Drilling Program Site U1403 and U1408
sediments in the northwest Atlantic. ~~Earth and Planetary Sciences Letters~~ 584,
117518 (2022).
- 20. M.J. Jackson, B. Moskowitz, On the distribution of Verwey transition temperatures
in natural magnetites, ~~Geophysical Journal International~~ 224, 1314-1325 (2020).
- 21. L. Chang, A.P. Roberts, M. Winklhofer, D. Heslop, M.J. Dekkers, W. Krijgsman,
262 J.D. ^{fitz} Gerald, P. Smith, Magnetic detection and characterization of biogenic
magnetic minerals: ^A comparison of ferromagnetic resonance and first-order
reversal curve diagrams. ~~Journal of Geophysical Research: Solid Earth~~ 119, 6136-
6158 (2014).
- 22. Expedition 342 Scientists. Site U1408. In Norris, R.D., Wilson, P.A., Blum, P., and
the Expedition 342 Scientists, Proc. IODP, 342: College Station, TX (Integrated
Ocean Drilling Program) (2014).
- 23. Y. Yamamoto, H. Fukami, W. Taniguchi, P.C. Lippert, Data report: updated
magnetostratigraphy for IODP Sites U1403, U1408, U1409, and U1410. In Norris,
R.D., Wilson, P.A., Blum, P., and the Expedition 342 Scientists, Proc. IODP, 342:
College Station, TX (Integrated Ocean Drilling Program) (2018).
- 24. J.P. Valet, L. Meynadier, Y. Guyodo, Geomagnetic dipole strength and reversal rate

- over the past two million years. Nature 435, 802-805 (2005). *un-dash*
- 25. L.B. Ziegler, C.G. Constable, C.L. Johnson, L. Tauxe, PADM2M: a penalized
maximum likelihood model of the 0-2 Ma palaeomagnetic axial dipole moment.
Geophys. J. Int. 184, 1069-1089 (2011).
- 26. Y. Yamamoto, T. Yamazaki, G.D. Acton, C. Richter, E.P. Guidry, C. Ohneiser,
Palaeomagnetic study of IODP Sites U1331 and U1332 in the equatorial Pacific -
extending relative geomagnetic palaeointensity observations through the Oligocene
and into the Eocene. Geophys. J. Int. 196, 694-711 (2014).
- 27. T. Yamazaki, H. Oda, A Brunhes-Matuyama polarity transition record from anoxic
sediments in the South Atlantic (Ocean Drilling Program Hole 1082C). Earth
Planets Space 53, 817-827 (2001).
- 28. P.L. Olson, R.S. Coe, P.E. Driscoll, G.A. Glatzmaier, P.H. Roberts, Geodynamo
reversal frequency and heterogeneous core-mantle boundary heat flow. Phys. Earth
Planet. Inter., 180, 66-79 (2010).
- 29. P. Olson, H. Amit, Magnetic reversal frequency scaling in dynamos with
thermochemical convection. Phys. Earth Planet. Inter., 229, 122-133 (2014).
- 30. K. Ohta, Y. Kuwayama, K. Hirose, K. Shimizu, Y. Oishi, Experimental
determination of the electrical resistivity of iron at Earth's core conditions. Nature

- 534, 95-97, doi:10.1038/nature17957 (2016). an-dash
- 31. P.H. Roberts, E.M. King, On the genesis of the Earth's magnetism. Rep. Prog. Phys.
- 76, 096801 doi:10.1088/0034-4885/76/9/096801 (2013).
- 32. T. Nakagawa, C. J. Davies, Combined dynamical and morphological
- characterisation of geodynamo simulations. Earth Planet. Sci. Lett. 594, 117752,
- doi:10.1016/j.epsl.2022.117752 (2022).
- 33. Y. Lin, P. Marti, A. Jackson, Invariance of dynamo action in an early-Earth model.
- Nature 644, 109-114, doi:10.1038/s41586-025-09334-y (2025).
- 34. B. Buffett, Geomagnetic fluctuations reveal stable stratification at the top of the
- Earth's core. Nature, 507, 484-487 doi:10.1038/nature13122 (2014).
- 35. J. Si, R. Van der Voo, Too-low magnetic inclinations in Central Asia: An indication
- of a long-term tertiary non-dipole field? Terra Nova, 13, 471-478. (2001).
- 36. M. Westphal, Did a large departure from the geocentric axial dipole hypothesis
- occur during the Eocene? Evidence from the magnetic polar wander path of
- Eurasia. Earth and Planetary Science Letters, 117, 17-28 (1993).
- 37. J.G. Meert, N.M. Levashova, M.L. Bazhenov, E. Landing, Rapid changes of
- magnetic field polarity in the late Ediacaran: Linking the Cambrian evolutionary
- radiation and increased UV-B radiation. Gondwana Research, 34, 149-157 (2016).

38. W. Huang, J.A. Tarduno, T. Zhou, M. Ibañez-Mejia, L.D. Olmo-Barbosa, E. Koester,
E.G. Blackman, A.V. Smirnov, G. Ahrendt, R.D. Cottrell, K.P. Kodama, R.K. Bono,
D.G. Sibeck, Y.X. Li, F. Nimmo, S. Xiao, M.K. Watkeys. Near-collapse of the
geomagnetic field may have contributed to atmospheric oxygenation and animal
radiation in the Ediacaran Period. ~~Communications of Earth & Environment~~ 5,
207 (2024).

For figure caption edits, see ~~the~~
the figures.

Pp. 20-22 not scanned.

Fig. 1. Integrated stratigraphic data for IODP Site U1408¹⁸. (a) Litho-magnetostratigraphic and scanning X-ray-fluorescence-measured Ca/Fe ratio data. Rhythmic lithological alternations expressed in the sediment color, where the dark color corresponds to clay-rich sediments, and the light color to carbonate-rich sediments (see also 'b' panel). The elementary lithological cyclicality (obliquity, ~40 kyr) is bundled by a longer cyclicity (~173 kyr) shown by vertical, dashed lines and detected by the amplitude modulation (AM) method and the amplitude spectrogram, both applied to highly resolved (2 cm) XRF Ca/Fe ratio data. The spectral line in the amplitude spectrogram detects the elementary lithological cyclicality, and bifurcations within such spectral line (indicated by vertical arrows) track the longer ~173 kyr cyclicity. (b) Expanded view of the studied interval showing XRF Ca/Fe and color reflectance L* data, along with Gauss bandpass (1.1±0.5 cycles/m) filter outputs to extract the elementary cyclicity.

represents

of which were

of intervals
the

Fig. 2. Composite VGP and RPI record with interpreted geomagnetic polarity chrons. Downcore variations of the virtual geomagnetic pole (VGP) latitude (a), geomagnetic polarity chrons (b), and relative paleointensity (RPI) of the geomagnetic field (c) from an approximately 8 m composite interval of sediments at IODP Site U1408.

Core breaks? Connections between parts of the composite section?

Fig. 3. Reversal path phases for Chron 18n.1r and age variations of reversals. Classifications of precursor (blue), transition (red), and rebound (green) are based on the ~~previous~~ literature⁹ and age variations of reversals based on the astronomically-tuned age model for Site U1408¹⁸. Geomagnetic polarity chrons (a), the VGP latitude (b), and the RPI (c). Transitions are characterized by low RPI intervals (red horizontal lines in the RPI record).

This is a better word

of transitions

Average Duration of Polarity Reversal Events [x 40 kyr]

Fig. 4. Average durations of polarity reversal events from our numerical dynamo models. The horizontal axis shows dynamo simulation cases R1 to R8, which vary in their parameter values (Ekman and Rayleigh numbers), whereas the vertical axis shows the duration of each case; this value can be multiplied by 40 kyr to convert to approximate Earth years. The range of duration is represented by the 2-sigma-level intervals.

are indicated on the horizontal axis

are indicated on the horizontal axis

should

**METHODS**

Delete

**Samples**

Delete

Discrete samples were taken onboard the R/V JOIDES Resolution during IODP

Expedition 342 in 2012. Non-magnetic plastic cubes with a volume of 7 cm³ were

continuously inserted into the central part of the working half cores with ~2.2 cm

spacing. In total, 191 samples were collected from Hole A between 46.81 ^{and} 51.45 m CFS

(Core depth below ^{srf} Sea Floor) and 262 samples were taken from Hole C between

49.82 ^{and} 55.78 m CFS. A between-hole composite depth scale in meters CCSF (Core

366 ^{srf} Composite depth below ^{srf} Sea Floor) was constructed based on between-hole correlation

of physical, chemical, and magnetic properties^{18,22}. Mid-point depths of the samples

correspond to 50.17-54.80 m CCFS (Hole A) and 52.18-58.13 m CCFS (Hole C).

370 ^a **Paleo- and rock magnetic measurements** ^g

Delete

Each sample was first analyzed by measuring its natural remanent magnetization

(NRM) and then was stepwise demagnetized using alternating field (AF) techniques to a

peak field of 80 mT, typically involving 21 demagnetization steps. An anhysteretic
remanent magnetization (ARM) was then imparted ~~to~~ ^{to} each sample ~~by~~ ^{with} a biasing DC
field of 100 μ T ~~with~~ ^{and} a peak AF of 80 mT ~~and~~ ^{was} then measured, followed by stepwise AF
demagnetization of ~~the~~ ^{the} ARM by a peak field of 80 mT. An isothermal remanent
magnetization (IRM) was then imparted ~~to~~ ^{to} each sample using a pulse ~~of~~ ^d field of 2.5 T; we
regarded ~~this~~ ^{for} value as the saturation IRM (SIRM) ~~of~~ ^{for} each sample. This SIRM was then
~~cleaned using~~ ^{subjected to} stepwise AF demagnetization to a peak field of 80 mT. Measurements of
the remanences and stepwise AF demagnetizations were made ~~by~~ using a 2G
Enterprises model 760R cryogenic magnetometer system with an inline static AF
demagnetizer and a Natsuhara-Giken model ASPIN-A spinner magnetometer. ~~The~~ IRMs
were imparted using a Magnetic Measurements model MPPM-10 pulse magnetizer.

After the remanence measurements, some ~~of the~~ samples were freeze-dried to conduct
low-temperature magnetometry using a magnetic property measurement system
(Quantum Design model MPMS-XL5) and first-order reversal curve (FORC)
~~measurements (Elliott et al., 1999)~~ ^{using} an alternating gradient magnetometer (Lake Shore model
MicroMag 2900 AGM). Small fractions ^{(a} few tens of mg) ^{of} ~~the~~ samples []] were used for
the low-temperature magnetometry: the fraction was first cooled from 300 to 5 K in a

field of 3 T (3T-FC-remanence) and temperature variation of the 3T-FC-remanence was
monitored from 5 to 300 K in zero field in 1-1.5 K steps. The fraction was again cooled
from 300 to 5 K in zero field and then an IRM was imparted in a field of 3 T
(3T-ZFC-remanence), followed by ~~monitoring~~ ^{measurement of the} temperature variation of the
3T-ZFC-remanence from 5 to 300 K in zero field. An IRM was further imparted for the
fraction in a field of 3 T at 300 K (3T-SIRM) and ~~the~~ variation of the 3T-SIRM was
~~monitored for a temperature cycle between 300 and 5 K. The~~ FORC measurements were
performed on another small ~~fractions of the~~ ^{measured} samples using the following measurement
parameters: 263 FORCs; field increment of 1 mT; local interaction fields (H_u) between
401 -50 and 50 mT; coercivity (H_c) from 0 to 150 mT; maximum applied field of 250 mT;
averaging time of 100-150 ms for each data point. FORC diagrams ^(Pike et al., 1999) were produced using
the FORCinel software with a VARIFORC algorithm to improve the smoothing of the
signal^{39,40}.

It is correct to cite the software & processing parameters
but you should also always cite Pike et al. - and any papers
that provide the interpretive framework used, and intellectual effort has gone
into that.

~~_____ Delete~~

408 Stepwise AF demagnetization results of NRM were used to calculate characteristic
remanent magnetization (ChRM) directions of the samples using principal component

Heslop & Roberts (2016) argued that anchoring is not well justified & provided a metric for whether fits can be considered to pass through the origin.

analysis (PCA) with a fit anchored to the origin⁴¹. Except for a few samples, ChRM
~~directions~~ ^{fits} have maximum angular deviation (MAD) values less than 10² degrees
(Extended Data Fig. 1). The cores were recovered from the seafloor using the FlexIt
core orientation tool, which provides the magnetic tool face (MTF) orientation value
giving the angle between geomagnetic north and the double line on the core liner for
each core⁴². Thus, the relative declination of the ChRM direction ~~of~~ ^{for} each sample is
transferred into a geographic coordinate to calculate a virtual geomagnetic pole (VGP)
latitude.

Determination of relative paleointensity (RPI) of the geomagnetic field.

420 ^a
Delete

These figures are cited before Fig. 2 - must be called out in order.

Stepwise AF demagnetization results of NRM, ARM, and IRM were used to construct
diagrams of NRM-ARM and NRM-IRM for each sample. Best-fit slopes ~~of the~~
~~diagrams~~ were individually determined, giving values of the slopes as $slope_{NRM-ARM}$ and
$slope_{NRM-IRM}$ (Extended Data Figs. 3 and 4). Overall averages of $slope_{NRM-ARM}$ were
calculated separately for ~~the~~ samples ^{from} of Hole A ($average_{NRM-ARM, Hole A}$) and Hole C
($average_{NRM-ARM, Hole A}$). They were used to normalize $slope_{NRM-ARM}$ for each sample ^{from}
Hole A ($slope_{NRM-ARM} / average_{NRM-ARM, Hole A}$) and Hole C ($slope_{NRM-ARM} / average_{NRM-ARM, Hole C}$).

Hole C). The same calculations were made on $slope_{NRM-IRM}$ ($slope_{NRM-IRM} / average_{NRM-IRM}$,
 Hole A; $slope_{NRM-IRM} / average_{NRM-IRM, Hole C}$). Relative paleointensity (RPI) of the
 geomagnetic field was ~~calculated by~~ ^{estimated as} an average of $slope_{NRM-IRM} / average_{NRM-IRM, Hole A}$
 and $slope_{NRM-IRM} / average_{NRM-IRM, Hole A}$ for the individual sample ^{from} Hole A, and by an
 average of $slope_{NRM-IRM} / average_{NRM-IRM, Hole C}$ and $slope_{NRM-IRM} / average_{NRM-IRM, Hole C}$
 for the individual sample ^{from} Hole C. Some samples ^{have} showed inhomogeneity ^{ous} in the rock
 magnetic property represented by ^{is} a ratio of ARM to SIRM (ARM/SIRM): RPI ^{for} of the
 samples ^{with} showing ARM/SIRM deviating from the averages (0.184 for Hole A; 0.180 for
 Hole C) by a one standard deviation ^{or more} were discarded.

 **Downcore VGP and RPI variations in the common depth scale.**

439  ^{Deletes}
 The downcore VGP and RPI variations ^{are plotted on} against the between-hole composite depth scale
 in meter ^{are evident} CCSF, ~~show~~ cm-scale offsets between Hole A and Hole C for the overlapping ^{ing}
 interval (~52.2-54.8 m CCFS). ^C ~~Because the~~ calibration of the geochronology was
 mainly made ^{using} on the X-Ray fluorescence-derived Ca/Fe ^{record from} ratio of Hole C ^W ¹⁸ we held the
 Hole C depths fixed (CCFS_{Hole C}) and shifted the Hole A depth (CCFS_{Hole A}) with respect
 to the Hole C depth to match the variation between Hole A and Hole C. We found that ^a

4-cm upward-shift ($CCFS_{\text{Hole A}} - 0.04 \text{ m}$) ~~results in~~ ^{produces} the best match ~~of the~~ ^{for} variations
 between the two holes, ~~and thus~~ ^{so} the Hole A depth was correlated to the Hole C depth
 ($CCFS_{\text{Hole C}}$) in this way (Extended Data Fig. 5). RPI values from Hole C were rescaled
 to have the same average as those of Hole A for the overlapp~~ed~~ ^{ing} interval. The composite
 VGP and RPI variations were adopted from the interval ~~of~~ 50.13-53.12 m $CCFS_{\text{Hole C}}$ ~~in~~
 ~~the~~ Hole A and that of 53.12-58.13 m $CCFS_{\text{Hole C}}$ in ~~the~~ Hole C.

**Numerical geodynamo simulations**

Delete

We solve the equations for chemically-driven convection of the Boussinesq fluid,
 transport of the light element concentration, and magnetic induction in a spherical shell
 rotating with an angular rotation rate, Ω ^{43,44}. The ratio of the inner core radius (r_i) to
 that of the outer core (r_o) is set to $r_i/r_o = 0.35$. The inner core and the mantle are
 assumed to be insulating. Boundary conditions are no-slip for the velocity field, and
 fixed flux for the light element concentration. We set the ratios between the fluid
 viscosity ν , the compositional diffusivity κ^C and the magnetic diffusivity η to give the
 compositional Prandtl number $Pr^C = \nu/\kappa^C = 1$ ^{and} the magnetic Prandtl number $Pm =$
 $\nu/\eta = 20$. The Ekman number is set to $Ek = \nu/2\Omega D^2 = 3.25 \times 10^{-3}$ ^{and} 2×10^{-3}

Do you cite ED fig. 6?

(here $D = r_o - r_i$ is thickness of the shell). The Rayleigh number $Ra = \alpha g \varepsilon D^3 /$
$2\Omega\nu\kappa^C$ (here α is the rate of compositional expansion, g is the gravitational acceleration
at the core-mantle boundary, and ε is the uniformly distributed volumetric sink of light
elements, respectively) is varied to obtain dynamo solutions.

We simulated eight dynamos with different values of Ek and Ra , these are named R1 –
R8, respectively. The start of a reversal in ~~direction~~ is defined as the instance when the
dipole axis crosses the latitude of $\pm 45^\circ$ toward the equator ~~before~~ polarity change.
Similarly, the end of a reversal in ~~direction~~ is defined as the instance when the dipole
axis finally crosses the latitude of $\pm 45^\circ$ toward the geographic pole after crossing the
equator. Stability of the dipole axis after crossing the equator for at least one dipole free
decay time is required to distinguish reversals from short events such as excursions^{29,34}.

476 ^{Then} Mean and standard deviation of the sampled reversal durations in cases R1 – R8 are
477 calculated according to a lognormal distribution¹³. Our definition of a ~~reversal~~ ^{directional}
~~direction~~ is conservative, which results in shorter estimated durations than other
definitions. Time is scaled using the dipole free decay time $\tau_{dip} = r_o^2 / (\pi^2 \eta)$, or ~ 40
480 kyr using a conservative estimate of $\eta = 1 \text{ m}^2/\text{s}$ ⁴⁵. The latest estimate of higher
electrical conductivity yields a longer dipole free decay time³⁰. Properties of the

dynamo solutions are summarized in Extended Data Table 1.

**Data availability**

The data that support the findings of this study are available in the extended data table
(key result values of numerical geodynamo simulations) and via Zenodo ~~with the~~
~~identifier~~ (10.5281/zenodo.13917223) which will be publicly accessible upon
publication. Source data are provided with this paper.

**Code availability**

The codes used during the current study are available from the author on reasonable
request.

*This does not meet journal requirements.
Code must be made publicly available.*

**Methods references**

39. R.J. Harrison, J.M. Feinberg, FORCinel: ^a An improved algorithm for calculating
first-order reversal curve distributions using locally weighted regression smoothing.
Geochem., Geophys., Geosyst. 9, Q05016. doi .org /10 .1029 /2008GC001987

(2008).

40. R. Egli, VARIFORC: An optimized protocol for calculating non-regular first-order reversal curve (FORC) diagrams. *Glob. Planet. Change* 110 302-320 (2013).

41. J.L. Kirschvink, The least-squares line and plane and the analysis of paleomagnetic data. *Geophys. J. R. astr. Soc.* 62, 699-718 (1980).

42. Expedition 342 Scientists. Methods. In Norris, R.D., Wilson, P.A., Blum, P., and the Expedition 342 Scientists, *Proc. IODP, 342: College Station, TX (Integrated Ocean Drilling Program)* (2014).

43. F. Takahashi, Implementation of a high-order combined compact difference scheme in problems of thermally driven convection and dynamo in rotating spherical shells. *Geophys. Astrophys. Fluid Dyn.* 106, 231-249 (2012).

44. F. Takahashi, Double diffusive convection in the Earth's core and the morphology of the geomagnetic field. *Phys. Earth Planet. Inter.* 226, 83-87 (2014).

45. D. Gubbins, P.H. Roberts, *Magnetohydrodynamics of the Earth's core*. In *Geomagnetism Vol. 2* (Ed. Jacobs, J. A.), 1-183 Academic Press, London (1987).

Use an en-dash for consistency

a

↑

#

**Acknowledgements**

Delete

This research used samples and data provided by the Integrated Ocean Drilling Program
(IODP). We ^{thank} ~~are indebted to~~ the staff of the R/V JOIDES Resolution ^{and} ~~and~~ the Bremen
Core Repository, ^{and} ~~We also gratefully acknowledge~~ the Expedition 342 science party. This
research was supported through the Japan Agency for Marine-Earth Science and
Technology (JAMSTEC) IODP Expedition 342 After Cruise Research Program, the
Japan Society for the Promotion of Science (JSPS) KAKENHI (JP15H05832,
JP16H04043, JP18K03808, JP21K03725 and JP24K07119), and the Kochi University
Research Project (Earth Investigation Project: Past, present, and future of environment,
earthquake, and resources recorded in the ocean and land). S.B. was supported by the
French Agence Nationale de la Recherche (19-CE31-0002 AstroMeso) and the
European Research Council under the European Union's Horizon 2020 Research and
Innovation Program (Advanced Grant AstroGeo-885250). ~~The~~ ^C computation was carried
out using ~~the~~ computer facilities at the Research Institute for Information Technology,
Kyushu University.

**Author contributions**

Delete

Y.Y., S.B. and P.C.L. designed the project.

Y.Y. conducted paleo- and rock magnetic measurements and analyzed the results.

S.B. developed the geochronology model.

F.T. conducted numerical geodynamo simulation⁵ and analyzed the results.

All authors contributed to ~~the~~ discussion of ~~the~~ results and editing of the manuscript.

**Competing interest declaration**

Delete

The authors declare no competing interests.

**Additional information**

Delete

Correspondence and requests for materials should be addressed to Y. Yamamoto.

for edits to figure captions, see figures.
Pages 34-36 not scanned.

Add degree symbols in each case.

Extended Data Fig. 1

Representative orthogonal vector plots of the AF demagnetization results from NRM1 of the samples. Solid (open) circles indicate horizontal (vertical) projections. Gray circles stand for data not used in PCA analysis. Black dashed lines are best-fit to the data.

These MADs are inflated by anchoring - see separate comment.

Extended Data Fig. 2

Representative results of the low-temperature magnetometry on the sample fractions selected from three horizons (a-c) and those of the FORC diagrams from the same horizons (d-f).

Extended Data Fig. 3

Representative NRM-ARM diagrams for the AF demagnetization results of the samples. Best-fit slopes are determined for the diagrams and their values with the correlation coefficients (r) and the number of data points (N) are indicated.

fan studied

Extended Data Fig. 4

Representative NRM-IRM diagrams for the AF demagnetization results of the samples. Best-fit slopes are determined for the diagrams and their values with the correlation coefficient (r) and the number of data points (N) are indicated.

from studied

Extended Data Fig 5

Downcore VGP and RPI variations ^{plotted on} the common depth scale. The between-hole composite depth scale in meter CCSF was further adjusted to have the best matches in the variations between Hole A (blue) and Hole C (red). ^{the} Hole A depth ^{was} correlated to ^{the} Hole C depth (CCFS_{Hole C}) by shifting the Hole A record ^{up} 4 cm upward (CCFS_{Hole A} - 0.04 m).

Extended Data Fig. 6.

Typical examples of time-series for (a) simple and (b) complex polarity reversals in a numerical dynamo model (case-R2). Tilt angle represents deviations of the dipole from the rotation axis in degree. Denoted in green is the durations of the reversal events according to the criteria adopted in the present study.

two

Extended Data Table 1

Results of numerical geodynamo modeling. Run ID: An ID code representing each dynamo case; Ek : Ekman number; Ra : Rayleigh number; Total run time: simulation run time in the unit of dipole free decay time; Mean duration: mean of the sampled reversal duration; Range of durations: minimum and maximum values of the reversal duration; Rm : Magnetic Reynolds number in terms of volume-averaged velocity; Λ : Elsasser number in terms of volume-averaged magnetic field.

Run ID	Ek	Ra	Total run time	Number of reversals	Mean duration	Range of durations	Rm	Λ
R1	0.00325	100	146.0	20	0.214	[0.045, 1.016]	167	8.8
R2	0.00325	105	156.4	25	0.300	[0.062, 1.441]	172	9.5
R3	0.00325	110	166.8	22	0.311	[0.061, 1.598]	178	9.4
R4	0.00325	119.275	125.1	22	0.644	[0.126, 3.298]	194	8.1
R5	0.002	180	127.3	15	0.194	[0.044, 0.855]	300	19.6
R6	0.002	200	127.5	15	0.182	[0.040, 0.822]	326	17.2
R7	0.002	220	135.5	20	0.275	[0.054, 1.393]	339	20.6
R8	0.002	250	90.1	15	0.227	[0.045, 1.156]	372	17.8